# Dietary Polysaccharides Exert Anti-Fatigue Functions via the Gut-Muscle Axis: Advances and Prospectives

**DOI:** 10.3390/foods12163083

**Published:** 2023-08-17

**Authors:** Yaping Zhou, Zhongxing Chu, Yi Luo, Feiyan Yang, Fuliang Cao, Feijun Luo, Qinlu Lin

**Affiliations:** 1National Engineering Research Center of Deep Process of Rice and Byproducts, Hunan Key Laboratory of Grain-Oil Deep Process and Quality Control, College of Food Science and Engineering, Central South University of Forestry and Technology, No. 498, Shaoshan Road, Changsha 410004, China; zyp4265@163.com (Y.Z.); czxcs2021@163.com (Z.C.); yfyei410@163.com (F.Y.); t20121480@csuft.edu.cn (F.L.); 2Department of Clinical Medicine, Medical College of Xiangya, Central South University, Changsha 410008, China; lyzndx0810@163.com; 3Co-Innovation Center for the Sustainable Forestry in Southern China, College of Forestry, Nanjing Forestry University, Nanjing 210037, China; fuliangcaonjfu@163.com

**Keywords:** dietary polysaccharides, fatigue, action mechanism, gut microbiota, gut-muscle axis

## Abstract

Due to today’s fast-paced lifestyle, most people are in a state of sub-health and face “unexplained fatigue”, which can seriously affect their health, work efficiency, and quality of life. Fatigue is also a common symptom of several serious diseases such as Parkinson’s, Alzheimer’s, cancer, etc. However, the contributing mechanisms are not clear, and there are currently no official recommendations for the treatment of fatigue. Some dietary polysaccharides are often used as health care supplements; these have been reported to have specific anti-fatigue effects, with minor side effects and rich pharmacological activities. Dietary polysaccharides can be activated during food processing or during gastrointestinal transit, exerting unique effects. This review aims to comprehensively summarize and evaluate the latest advances in the biological processes of exercise-induced fatigue, to understand dietary polysaccharides and their possible molecular mechanisms in alleviating exercise-induced fatigue, and to systematically elaborate the roles of gut microbiota and the gut-muscle axis in this process. From the perspective of the gut-muscle axis, investigating the relationship between polysaccharides and fatigue will enhance our understanding of fatigue and may lead to a significant breakthrough regarding the molecular mechanism of fatigue. This paper will provide new perspectives for further research into the use of polysaccharides in food science and food nutrition, which could help develop potential anti-fatigue agents and open up novel therapies for sub-health conditions.

## 1. Introduction

Fatigue is a subjective feeling of discomfort which can be described as an overwhelming feeling of tiredness and exhaustion, and which occurs in the context of various physiological, pathological, and psychological imbalances of organisms [1]. Fatigue, as a common problem in modern societies worldwide, must be taken seriously. Physical labor, mental stress, altitude hypoxia, and severe or long-term illness can cause fatigue in the body [2]. Simultaneously, as a complex and multifactorial symptom, fatigue also harms the body in all aspects. The tremendous consequences of fatigue are consistent across physiological, pathological, and psychological diseases [3]. During long-term high-intensity exercise, the internal environment of the body is disturbed due to energy consumption and metabolite accumulation, which may lead to muscle soreness and spasm, dyskinesia, etc. [4]. In severe cases, fatigue may lead to altered endocrine function, impaired immunity, systemic inflammation, and organic diseases with health threats, such as cancer, aging, diabetes, etc. [5,6,7]. Additionally, fatigue can also contribute to anxiety and depression, along with the onset of neurological disease [8]. Fatigue is a major health problem that affects individual voluntary activities or disease recovery. Particularly, exercise-induced fatigue seriously affects the daily lives of people [9]. Therefore, it is crucial to find ways to control and reduce fatigue. Interestingly, studies and applications of natural polysaccharides in relieving exercise-induced fatigue have garnered interest due to their good curative results and few side effects.

Polysaccharides derived from natural products are valued for effective and extensive health-related activities [10,11]. Polysaccharide is one of the crucial building blocks of cell structures; it is a biopolymer composed of many monosaccharides connected by glycosidic bonds, exhibiting structural diversity and complexity, no sweetness, and insolubility in water [12]. Several reviews have suggested that polysaccharides are natural and effective substances that exert antifatigue activities [3,9,13]. Natural polysaccharides can exert anti-fatigue activity in different manners, and the underlying molecular mechanisms are mainly related to improving energy metabolism [14], eliminating excess metabolites [15], enhancing mitochondrial biogenesis [16], enhancing immune function [17], inhibiting oxidative stress and inflammatory response [18], regulating neurotransmitters [19], etc. At the same time, these mechanisms also interact and exert mutual regulation. When polysaccharides exert anti-fatigue effects, they can usually improve exercise endurance or reduce muscle damage [15,16]. Emerging data indicate that polysaccharides play extensive roles in host–gut microbiota symbiosis. It is well known that gut microbes digest various dietary polysaccharides and produce oligosaccharides, which can selectively promote the growth of beneficial bacteria and are fermented into short-chain fatty acids (SCFAs) that interact closely with intestinal cells, liberating host-absorbable energy [20]. Recent studies have found that polysaccharides can play an anti-fatigue role by regulating gut microbiota [21,22]. In addition, evidence suggests that the composition and diversity of gut microbiota may be determinants of skeletal muscle metabolism and functionality [23]. Therefore, it is possible that some natural polysaccharides play an active role in fatigue by mediating the gut-muscle axis.

The gut-muscle axis is a newly proposed concept in the field of sports medicine and nutrition. Recent investigations have shown that muscle function and metabolism largely depend on the number and composition of the gut microbiota, which holds promise as a potential intervention target for the prevention and treatment of muscle-related diseases [24]. The abundance of important components of human fecal microbiota, such as *Enterobacteriaceae*, *Bacteroides*, and *Prevotella*, has been correlated with measures of muscle fitness [25]. Even though the concept of the gut-muscle axis has been proposed, its causal relationship is still unclear. Previous investigation has shown that there is a relationship between sarcopenia and gut microbiota, known as the gut-muscle axis [26,27]. A previous review summarized that gut microbiota can interact with the host’s muscle mass and function through regulating inflammation, immunity, and energy metabolism, directly or indirectly establishing a connection with muscle disease, thereby realizing the “gut-muscle axis” and affecting the physiological function of the body [28]. Although some studies have revealed the manner in which polysaccharides influence the gut microbiota to resist exercise fatigue [29,30] and affect muscle quality [31,32], there are few reports on how polysaccharides improve exercise-induced fatigue via the gut-muscle axis. Further research is needed to explore the potential therapeutic effects of polysaccharides on exercise-induced fatigue and the gut-muscle axis.

## 2. Pathogenesis of Exercise-Induced Fatigue

In kinematics, fatigue is a multidimensional concept which is of great significance to performance during exercise [33]. Exercise-induced fatigue is non-pathological fatigue, which is described as a decrease in physical (muscle) strength and/or mental state that comes from intense and prolonged exercise [9,34]. At present, its complex multifactorial mechanisms are not completely clear; however, the peripheral and central theories are generally accepted as the two major paths of exercise-induced fatigue [35]. Fatigue during exercise can be approached from different angles.

### 2.1. Pathogenesis of Peripheral Fatigue

Peripheral fatigue is usually described as an impairment located in the muscle. The peripheral theory of exercise-induced fatigue mainly includes energy exhaustion, metabolite accumulation, immunoregulation, oxidative stress, inflammation, etc. The energy metabolism in muscles determines the degree of exercise-induced fatigue [36]. The occurrence of muscle fatigue may mediate insufficient adenosine triphosphate (ATP) production and imbalance the release of Ca^2+^ from the sarcoplasmic reticulum Ca^2+^-ATPase pump [37]. Furthermore, the dysfunction of ATP production and utilization will lead to the accumulation of metabolic by-products [38]. The metabolite accumulation theory holds that excess metabolites generated during intense exercise can accumulate in skeletal muscles and blood, hence interrupting homeostasis and impacting blood lactic acid (BLA), malondialdehyde (MDA), lactate dehydrogenase (LDH), etc. [9,15], causing fatigue.

Additionally, energy expenditure is reserved for the immune system, and exercise is related to physical improvement and immune adaptations [39]. Fatigue is initially an adaptive immune-mediated response of the body to improve the production of resistance/anti-inflammatory cytokines and contribute to disease resistance [40]. Many studies have searched for evidence of compromised immune systems, such as T or B lymphocyte proliferation and cytokine overproduction [17,41], in fatigue patients. Low levels of reactive oxygen species (ROS) are necessary for the generation of normal muscle strength, but high concentrations of ROS can lead to impaired muscle contractility [42]. Studies indicated that extended endurance exercise could produce large amounts of ROS, which affected the exercise ability of muscles and caused muscle fatigue [43]. Elevated ROS will not only lead to an imbalance of the redox system in the body, but may also trigger the lipid peroxidation of the mitochondrial membrane and disrupt mitochondrial functions, affecting energy supply and ultimately leading to fatigue [44,45]. Additionally, excessive free radicals produced during exercise can lead to multiple cell/organelle damage and tissue/organ pathophysiological changes, thereby affecting exercise capacity [46,47].

Inflammation is thought to be another important molecular mechanism of fatigue. Acute exercise increases the number of circulating inflammatory cells and cytokines to maintain physiological homeostasis [39]. Studies have shown that exercise overload can promote an inflammatory profile, leading to an increase in pro-inflammatory cytokines such as interleukin-6 (IL-6), interleukin-1 beta (IL-1β), tumor necrosis factor-alpha (TNF-α), etc. [32,37]. In addition, high levels of oxidative stress may lead to the deterioration of inflammatory markers [43]. Excessive ROS seems to be associated with the mechanisms that limit aerobic exercise and may also cause acute and chronic inflammation, which in turn leads to decreased muscle strength and increased fatigue [39]. Zhu et al. showed that the peripheral fatigue index, including energy sources and metabolites, was significantly correlated with the levels of inflammatory factors and ROS [48]. Furthermore, fatigue is also related to diseases associated with systemic inflammation [49]. The sequence of peripheral theories leading to exercise-induced fatigue can be schematically denoted as: energy exhaustion → metabolite accumulation → immunoregulation → oxidative stress → inflammation.

### 2.2. Pathogenesis of Central Fatigue

Central fatigue is defined as a failure of the central nervous system (CNS) to adequately drive the muscle. Autonomic neuromodulation and endocrine regulation are involved in the central theory of exercise-induced fatigue [9]. Central fatigue involves the regulation mechanism of the cerebral cortex and the CNS and is related to the continuous stimulation of nerve excitation by skeletal muscle contractions [50]. Neurotransmitters determine and create communication between neurons, and their mediated defense effect plays an essential role in exercise-induced fatigue [33]. Within fatigue investigation, the greatest emphasis has been given to neurotransmitters released by neurons, including 5-hydroxtryptamine (5-HT), dopamine (DA), noradrenaline (NA), gamma-aminobutyric acid (GABA), and acetylcholine, which can alter central fatigue and motoneuronal output [50], according to the report that the enhancement of endogenous 5-HT concentration exacerbates central fatigue during sustained maximal contraction [51]. New investigations have shown that the release of 5-HT onto motoneurons was linked to the intensity of muscle contraction, and high levels of 5-HT damaged the voluntary activation of muscles in strong contraction-induced fatigue [52]. DA is the most abundant catecholamine neurotransmitter in the brain, and it regulates multiple physiological functions of the CNS [53]. Clinical evidence suggests that fatigue symptoms may be caused by impaired striatal dopaminergic neurotransmission [54]. After DA depletion, some striatal structures showed attenuated connectivity with the medial prefrontal cortex, which in turn exhibit exacerbated fatigue [55]. During strenuous exercise, DA is involved in the connectivity and structural integration of large-scale networks in the brain, and the neural states within these networks enter the brain as interoceptive information to regulate the development of fatigue [56]. In addition, DA can also manipulate the onset of fatigue by interfering with thermoregulation [57]. Notably, the noradrenergic neurotransmitter system accelerates central fatigue during prolonged exercise [33,58]. The GABA/glutamic acid pathway and acetylcholine participate in the regulation of cognitive and emotional disorders in central/mental fatigue [59,60]. Several neurotransmitter systems may be implicated in multiple brain regions, and the sum of these changes may explain the impairment of exercise endurance performance in a state of mental fatigue [33]. In general, brain neurochemistry (i.e., brain neurotransmitters) is involved in the complex regulation of fatigue.

Furthermore, exogenous forms of neuromodulation, such as non-invasive brain stimulation, may also cause fatigue [50]. The use of anodic transcranial direct current stimulation in the frontal brain regions may be a novel strategy to reduce mental fatigue and maintain endurance performance in swimmers [61]. Electrophysiological treatment (transcranial direct current stimulation and transcranial magnetic stimulation) participates in regulating fatigue symptoms in patients with multiple sclerosis by altering corticospinal excitability [62]. The direct current stimulation can cause transient or long-term changes in the excitability of cortical neuronal pathways and mitigate the occurrence and development of fatigue.

Endocrine disorders can also cause fatigue. Recent studies have shown that catecholamines in the brain are intimately connected with episodes of fatigue during endurance exercise [33]. The CNS controls the regulation of energy balance through signal transduction, including the release of leptin, adiponectin, ghrelin, etc., and these signal transduction molecules provide important feedback for the hypothalamus, enabling it to regulate energy homeostasis [43]. Moreover, the dysfunction of the hypothalamus–pituitary–adrenal (HPA) axis tends to amplify the fatigue and stress response of patients through various signaling pathways [63]. The regulation of endogenous cholesterol and testosterone synthesis can prevent and treat exercise-induced low serum testosterone levels and improve the exercise capacity of Sprague Dawley (SD) rats [64]. Likewise, changes in the muscle endurance and indexes of endocrine function, including decreased total testosterone readings and increased cortisol levels, were noted in handball players after acute high-intensity resistance exercise [65]. Moreover, the endocrinological evaluations found that fatigue symptoms of COVID-19 patients may mediate the regulation of the HPA axis, adrenal behavior, and thyroid function [66].

The central and peripheral theories are two partnering causes of fatigue. Obviously, it must be remembered that fatigue is a very complex construct and that, besides the peripheral theory and the central theory, there are several other factors that play a role in its occurrence [33]. Fatigue is affected by factors such as psychological, genetic, environmental, and gender influences [35]. Fatigue can occur for a variety of reasons, so the causes of fatigue and methods of intervention must be considered from multiple perspectives.

## 3. Polysaccharides Ameliorate Fatigue

Polysaccharide is a type of active biological macromolecule which widely exists in plants, animals, and microbes, characterized by high universality, abundance, and biological activity [10,20,67]. In recent years, natural polysaccharides have received growing attention and great interest for use in medications, health foods, and cosmetics [11]. Polysaccharides have been considered to be a new natural and effective anti-fatigue substance [9]. Some natural polysaccharides, such as *Semen cassiae* (*Cassia obtusifolia* L.) [14], *Ganoderma lucidum* [15], *Lycium barbarum* [68], *Cordyceps militaris* [69], and apple pomace [70], exert beneficial anti-fatigue effects via different pathways (Figure 1). The monosaccharide composition and the anti-fatigue mechanisms of natural polysaccharides are summarized in Table 1.

### 3.1. Impacts on Energy Metabolism

Energy sustains the various life activities of the body. The main energy substances necessary for muscle fiber contraction are ATP, glycogen, and fat [3]. The energy expenditure was based on different exercise degrees. Short-term exercise, moderate exercise, and long-term endurance exercise gradually deplete the stores of ATP, glycogen, and fat as their main energy supplies [9]. Exercise-induced fatigue will occur with the continuous consumption of ATP and an insufficient energy supply of glycogen and fat [4]. Carbohydrates are the primary consumed substrate during moderate exercise. In general, natural polysaccharides, with significant anti-fatigue effects, usually have the ability to regulate energy metabolism by increasing ATP content, increasing glycogen reserve, regulating lipid metabolism, and enhancing mitochondrial function.

In animal experiments, it was discovered that *Lycium barbarum* polysaccharide (LBP) [68] and *Pholiota nameko* polysaccharide (PNP) [80] could alleviate fatigue by increasing the level of ATP and Ca^2+^-Mg^2+^-ATPase in the body. Glycogens are the primary substrates for maintaining glycolysis, oxidative phosphorylation, and blood glucose levels in the body during endurance exercise [3,36]. In an experiment involving the forced swimming of mice, it was shown that LBP1-decorated selenium nanoparticles (LBP1-SeNPs) were able to mitigate fatigue by increasing the reserve of glycogen in the liver and muscles [79]. This result was consistent with most other findings [14,17,70]. Importantly, adenosine 5‘- monophosphate-activated protein kinase (AMPK) is a master regulator that senses the energy state, maintains glucose homeostasis, and mediates beneficial cellular adaptations in many vital tissues and organs [83]. AMPK triggers the catabolic pathways that produce ATP. Peroxisome proliferator-activated receptor-gamma coactivator-1alpha (PGC-1α) can be directly activated by phosphorylated AMPK (p-AMPK), promoting the conversion of muscle fibers to more effectively resist fatigue and produce more ATP, which provides energy for muscle contraction [84]. Natural LBP possesses marked anti-fatigue effects during exercise, which may be implemented by regulating energy metabolism through the AMPK/PGC-1α signaling pathway [68]. Further investigation demonstrated that *Panax ginseng C. A. Meyer* polysaccharide (APS-1) supplement could improve glucose uptake and regulate glycolysis in mice via the upregulated expression of AMPK axis-related proteins such as AMPK, p-AMPK, PGC-1α, and glucose transporter 4 (GLUT-4) in muscle tissues [78].

The availability of carbohydrates can affect fat metabolism and energy supply during exercise. Studies have proven that McArdle patients have a unique fat oxidation capacity, which may serve as an adaptation to compensate for the genetic blocking of glycogen metabolism [85]. Some natural polysaccharides delay glycogen consumption by promoting fat/lipid metabolism. The *Ganoderma lucidum* polysaccharide (GLPs), with molecular weights greater than 10 kDa, can prolong swimming time and improve endurance by accelerating fat conversion [15]. During exercise, AMPK is activated to promote fatty acid oxidation and accelerate glycogen absorption [83]. Peroxisome proliferator-activated receptor (PPAR) is a transcription factor that maintains lipid homeostasis. AMPK and PPAR signaling and fatty acid metabolism promote the regulation and role of glycogen phagocytosis in skeletal muscle metabolism [86]. When PGC-1α is activated and binds to PPARα, it can maintain oxidation balance and regulate lipid metabolism by targeting several genes [84]. LBP combined with aerobic exercise has been reported to be beneficial to hepatic lipid metabolism via the AMPK/PPARα/PGC-1α pathways [87]. Generally, fatty acid oxidation can promote the production of ATP energy. Previous studies have also reported mechanisms to achieve anti-fatigue effects by modulating fat-related pathways [14,75].

Mitochondria are traditionally known as the powerhouse of the cell, producing cell energy in the form of ATP [36,88]. Fatigue caused by repetitive exhaustive exercise may cause oxidative damage to the mitochondrial membrane [89]. Improving mitochondrial function during exercise may be another mechanism to reduce fatigue [90]. Some natural polysaccharides exert excellent anti-fatigue effects by enhancing mitochondrial biogenesis and repairing mitochondrial dysfunction. For instance, Maca (*Lepidium meyenii Walp.*) aqueous extract (ME) was demonstrated to upregulate NAD^+^/NADH and mitochondrial biogenesis and function, thereby exerting anti-fatigue function by preventing mitochondria-mediated muscle damage and oxidative stress [16]. Red ginseng extract can exert anti-fatigue effects by ameliorating mitochondrial dysfunction by rescuing the density and morphology of the skeletal muscle mitochondria and increasing mitochondrial biogenesis [91]. PGC-1α is a transcriptional activator that activates mitochondrial oxidative metabolism and biogenesis [3]. When PGC-1α is modulated by p-AMPK, it triggers the PGC-1α signaling cascade and induces mitochondrial biosynthesis [84]. Transmission electron microscopy showed that LBP increased the density of the mitochondria in rat liver [68]. APS-1 can ameliorate mitochondrial function and physiological performance by activating the AMPK-PGC-1α pathway to achieve beneficial anti-fatigue effects [78]. Moreover, red ginseng extract can prevent fatigue by activating the AMPK/PGC-1α cascade pathway, increasing the activity of Na^+^-K^+^-ATPase, and improving skeletal muscle mitochondrial function [91]. Thus, AMPK and PGC-1α are the key regulatory factors for polysaccharides to participate in energy metabolism. These investigations indicate that the anti-fatigue activity of natural polysaccharides is closely related to the energy metabolism mechanism of the body, which can increase glycogen reserves, promote fat/lipid catabolism, improve mitochondrial function, and provide energy for the body.

### 3.2. Reduce the Accumulation of Metabolites

Recovery from exercise-induced fatigue requires repairing the damage that has occurred in the body and/or promoting the elimination of metabolites accumulated during exercise [92]. Excessive metabolites (such as BLA, LDH, blood urea nitrogen (BUN), etc.) produced during strenuous exercise can accumulate in tissues and blood, thus disrupting homeostasis and leading to fatigue [9]. During sustained overtraining, anaerobic glycolysis accelerates with the accumulation of LA metabolites, resulting in changes in pH value, which often leads to the occurrence of fatigue [93]. The excessive accumulation of BLA and H^+^ can lead to acidosis, which affects the production and utilization of ATP [38]. The LDH-catalyzed reaction helps to generate sufficient ATP for exercise and accelerates the removal of LA, which delays ongoing metabolic acidosis [3]. It has been demonstrated that Maca polysaccharides (MP) could effectively elongate the exhausting swimming time of mice and decrease BLA, LDH, BUN, and liver glycogen (LG) levels to resist fatigue [74]. The polysaccharide fractions of *Inonotus obliquus* (PIO-1) showed the potential to improve fatigue by adjusting the activities of BLA, BUN, LDH, etc. [19]. In addition, the concentration of ammonia was increased by metabolism in skeletal muscles during intense exercise. Elevated ammonia could activate phosphofructokinase, inhibiting the oxidation of pyruvate to form acetyl CoA, which in turn promotes the production of LA, BUN, etc., thereby reducing endurance and causing fatigue [9]. The MP can clear the accumulation of LDH, BUN, and LA, as well as elongate the swimming duration and accelerate the average swimming speeds of mice [73]. Cai et al. found that the GLPs could increase fat transformation and decrease the activities of BUN, creatine kinase (CK), and BLA to promote fatigue recovery [15]. Additionally, CK is an important marker of tissue damage and a characteristic response to vigorous exercise. A polysaccharide-rich extract of *Phragmites rhizome* (PEP) markedly prolongs the swimming endurance capacity of mice, increases glucose levels, and decreases LDH and CK activities [75]. The water-soluble polysaccharide from *Semen cassiae* (*Cassia obtusifolia* L.) (SCP) exhibited strong anti-fatigue activity in mice; it can extend the weight-loaded swimming duration and decrease the levels of BUN, CK, triglyceride (TG), etc. in blood serum [14]. Studies have demonstrated that polysaccharides from corn silk [71], *Sarcodon imbricatus* (SI) [44], *Codonopsis pilosula* (*C. pilosula*) [76], and Chinese yam (*Dioscorea opposita Thunb.*) [18] also have anti-fatigue effects by regulating the accumulation of related metabolites.

### 3.3. Improve Immune Function

Certain polysaccharides are immunomodulators that play a central role in the regulation of the immune response during disease progression [12]. Under strenuous exercise, the immune system may become damaged, to some extent. Early clinical reports showed that the immune system of patients with fatigue syndrome was disturbed [40]. Some natural polysaccharides, such as *Dendrobium officinale* polysaccharide (DOP) extract, have also been shown to regulate their underlying immune function to improve fatigue, which can greatly increase the cell variability of T and B lymphocytes and alleviate fatigue syndrome caused by weight-loaded swimming [41]. Strikingly, the SCP exhibited anti-fatigue activity and an immunomodulatory effect in BALB/c mice; it can enhance the number of B-cell and T-cell lymphocytes in the spleen [14]. Steamed ginseng polysaccharides (WSGP-S3) can improve the exercise endurance and prolong the exhaustive swimming time; it can also delay exercise-induced fatigue and sports-related injuries by improving the biochemical indexes and enhancing the spleen cell proliferation (T or B lymphocyte) of trained mice [17]. In addition, chitosan oligosaccharide (COS), combined with running, promotes the development of the spleen and lungs, the level of lymphocytes, the ratio of T cell/CD8^+^ T cell, and improves the immune status of rats through cytokines such as TNF, interleukin-2 (IL-2), and interleukin-10 (IL-10), thereby mitigating fatigue [82]. Consequently, some polysaccharides that exhibit immunomodulatory effects on macrophages and lymphocytes, whose mechanism involves the neuroendocrine and immune system, may exert potential anti-fatigue activity. 

### 3.4. Enhance Antioxidant Activity

Oxidative stress is an important influence on fatigue. In fatigue caused by strenuous exercise, oxidative stress is primarily caused by an inadequate oxygen supply to organs and muscles, which can lead to unsustainable muscle tension and cause muscle contraction damage [42,46]. Free radicals are by-products of an organism’s metabolic processes, which exhibit dynamic equilibrium under normal circumstance. However, during excessive exercise, oxygen-containing free radicals will be produced and accumulated, which leads to the destruction of redox signaling or to molecular damage in the body [94]. The presence of antioxidant enzymes, such as catalase (CAT), glutathione peroxidase (GSH-Px), and superoxide dismutase (SOD), promotes oxidative defense, scavenges free radicals, and reduces oxidative damage [95]. In addition, as a major lipid peroxidation product, MDA can indicate the metabolic state of free radicals and indirectly reflect the extent of tissue damage [96]. Accordingly, some natural polysaccharides with antioxidant activity can eliminate fatigue by eliminating ROS, increasing antioxidant enzyme activities, and reducing MDA level. For example, the *Polygala tenuifolia* Willd. polysaccharide can decrease the concentrations of BLA and BUN and increase the levels of LG, muscle glycogen (MG), and LDH in exhaustive exercised mice; it exhibits high scavenging rates for hydroxyl free radical and DPPH free radical in vitro and exhibits good antioxidant properties [77]. It has been proven that APS-1 could reduce the accumulations of BLA, LDH, BUN, and MDA and increase the activities of SOD, CAT, and CK, which could prolong the fatigue tolerance of mice [78]. LBP was able to relieve fatigue by enhancing antioxidant enzyme levels and regulating metabolic mechanism [79]. Furthermore, it has been suggested that the ME can abrogate ROS accumulation, upregulate NAD^+^/NADH to reduce exercise-induced metabolic stress, and prevent oxidative stress-induced damage [16].

It is worth noting that AMPK, as an energy sensing factor, also supports redox equilibrium [83]. LBP plus aerobic exercise can regulate the synthesis and oxidation of liver fatty acids in SD rats by activating AMPK and increasing the expression of PPARα and its co-activator PGC-1α [87]. The activation of the AMPK/PPARα/PGC-1α signaling pathways can also target the regulation of the balance between oxidation and antioxidants, as well as reduce oxidative stress-related damage [84]. The anti-fatigue effect of APS-1 is related to its regulation of AMPK and PGC-1α in mouse muscle [78]. Furthermore, as a typical cellular antioxidant regulator at the transcriptional level, nuclear factor erythroid-derived 2-like 2 (Nrf2) can control the basic and induced expression of a series of antioxidant response element-dependent genes [69,97]. Studies have proven that the activation of the Kelch-like ECH-associated protein 1 (Keap1)/Nrf2/heme oxygenase-1 (HO-1) pathway is the dominant mechanism of cellular defense systems against oxidative stress, and it also participates in enhancing the exercise-induced fatigue state [47,94]. Usually, Nrf2 is present in the cytoplasm and binds to the inhibitory protein Keap1, but the activated Nrf2 can promote its separation from Keap1, subsequently binding to the antioxidant response element in the nucleus to control the transcriptions of downstream genes (SOD, CAT, and HO-1) and exacerbate its anti-oxidation ability [3,97]. Investigation indicated that SI exerts anti-fatigue activity by normalizing energy metabolism and Nrf2-mediated oxidative stress, including increasing the expression levels of Nrf2, SOD1, SOD2, HO-1, and CAT in the liver of CFS mice [44]. In similar research, the LBP could ameliorate exercise-induced oxidative stress by regulating the Nrf2/HO-1 pathway and energy metabolism via the AMPK/PGC-1α pathway [68]. Another study showed that *Cordyceps militaris* acidic polysaccharides (CMPB) can alleviate exercise fatigue and improve learning and memory of mice by stimulating the expression levels of phosphatidylinositol 3-kinase (PI3K), protein kinase B (AKT), Nrf2, and HO-1 proteins in the hippocampus [69]. Moreover, HO-1 can also inhibit the release of pro-inflammatory cytokines [3]. Consequently, the investigation and development of natural polysaccharides with the effect of improving oxidative stress and regulating AMPK and Nrf2-related pathways to improve fatigue has application value.

### 3.5. Inhibit Inflammatory Response

Prolonged and strenuous exercise causes acute and chronic inflammation, as well as the excessive release of pro-inflammatory cytokines (IL-1β, IL-6, and TNF-α), which further affect body function, leading to fatigue [37,94]. Accordingly, natural polysaccharides with anti-inflammatory activities may relieve fatigue. For example, COS can attenuate the activities of TNF-α and IL-2 in the serum of SD rats and improve the fatigue status [82]. Chinese yam polysaccharides (CYPs) can be used to combat fatigue by regulating the inflammatory pathways and oxidative stress, which can decrease the levels of IL-lβ, MDA, BUN, and LDH and increase the activity of ATP and SOD [18]. It has been proven that *Polygonatum sibiricum* polysaccharide (PSP) treatment not only markedly reduces the up-regulated of IL-1β, IL-6, and TNF-α, but also improves inflammation and glucose uptake in the L6 myotubes by regulating miR-340-3p/interleukin-1 receptor-associated kinase 3 (IRAK3) and GLUT-4 [32]. Furthermore, nuclear factor kappa B (NF-κB), as a transcription factor that activates the release of pro-inflammatory cytokines and generate a vicious circle of inflammatory response and mitochondrial dysfunction, as well as impaired mitochondria, produces large amounts of ROS, resulting in decreased muscle strength and fatigue [39,43]. Interestingly, astragalus polysaccharide (APS) has a protective effect on myocardial injury induced by excessive exercise in SD rat by activating the AMPK pathway, constricting the production of ROS, and reducing the levels of IL-1β, TNF-α, and NF-κB [98]. Additionally, interleukin-8 (IL-8) is also involved in mediating the development of inflammation. The yeast (*Saccharomyces cerevisiae*) beta-glucan (YBG) can inhibit the upregulation of macrophage inflammatory protein-1 beta (MIP-1β), IL-8, monocyte chemoattractant protein 1 (MCP-1), and TNF-α in serum, thus relieving the muscle injury and inflammation caused by exercise [99]. It is speculated that polysaccharide can inhibit the production of redundant cytokines, as well as attenuate inflammation and oxidative stress, thereby protecting the body and relieving fatigue.

### 3.6. Interfere with Autonomic Neuromodulation

It is generally believed that fatigue is also implicated in the central neurotransmitters (5-HT, DA, and NA) [33,50]. Excessive exercise can lead to physical fatigue, disrupt the balance of the oxidation/antioxidant system in the body, and thereby damage the CNS [69]. Investigations have shown that high concentrations of 5-HT can cause fatigue through motor neuron and muscle contraction [52]. The potential anti-fatigue activity of PIO-1 is partly link to a decreased concentration of 5-HT in mouse brains [19]. Polysaccharides from *Spirulina platensis* (PSP) can suppress the exercise-induced increase in 5-HT and tryptophan hydroxylase-2 (TPH2) and upregulate the expression of serotonergic type 1B (5-HT1B) to enhance exercise ability [81]. Furthermore, the interaction of 5-HT and DA plays a regulatory role in the development of exercise fatigue [54]. It has been confirmed that *Antrodia camphorata* polysaccharide (ACP) intervention could inhibit the expression of ROS-NLRP3 induced by 6-Hydroxydopamine (6-OHDA), protect the dopaminergic neurons, and improve the exercise capacity of mice [100]. Notably, *Herba Epimedii* polysaccharides (HEP2-a) exert ameliorative effects on metabolic disorders by increasing the level of NA and CK to treat fatigue [72]. Taken together, polysaccharides can mediate these neurotransmitters to alleviate the damage of the brain regions during exercise fatigue.

### 3.7. Regulate Endocrine System

Various studies have been searching for evidence of endocrine disruption, such as damage of secretory glands or endocrine cells, the disruption of hormone secretion, and the dysregulation of hypothalamic feedback regulation, in patients with fatigue, [9,33,43]. In addition, in long-term physical fatigue, the HPA axis was activated, regulating the synthesis and release of cholesterol and testosterone, potentially alleviating exercise-induced chronic pain, immunosuppression, and fatigue [64,65]. PEP exerts anti-stress and anti-fatigue effects by suppressing the over-activation of the HPA axis, as well as decreasing the content of TC and cortisol [75]. Thus, regulating endocrine disorders is considered another way for polysaccharides to relieve fatigue.

### 3.8. Other Anti-Fatigue Mechanisms

Studies over the past two years have also shown that some other natural polysaccharides show anti-fatigue properties or new anti-fatigue mechanisms. For example, they are involved in regulating changes in some related enzymes in cells [3,95]. PEP has showed a certain fatigue-relieving effect by maintaining the content of myeloperoxidase (MPO), GSH, SOD, CAT, GPx, and thiobarbituric acid-reactive substance (TBARS) [75]. Polysaccharide-rich extract from corn silk (PCS) can effectively improve fatigue, prolong swimming time in mice, and regulate the levels of enzymes related to liver function, such as alanine transaminase (ALT), aspartate transaminase (AST), and alkaline phosphatase (ALP) [71].

Moreover, polysaccharides may also regulate some other fatigue-related genes, such as GRAF1, BDNF, and LKB1. GRAF1 is a Rho-specific GTPase activating protein. It was proposed that PIO-1 can improve fatigue resistance by maintaining higher levels of GRAF1 expression in gastrocnemius muscles [19]. In particular, BDNF is a central regulator of energy homeostasis, and experiments have shown that CMPB can upregulate BDNF genes and exhibit anti-fatigue effects [69]. Additionally, APS-1 improves fatigue tolerance capacity via the upregulation of the expression of liver kinase B1 (LKB1) associated with the activation of AMPK [78]. Therefore, to some extent, alterations in these genes can also be used to evaluate fatigue.

Collectively, these conclusions indicate that the anti-fatigue effect of polysaccharides is complex and dynamically regulated (Figure 2). Polysaccharide extract from *C. pilosula* (POL) can efficiently enhance fatigue resistance by increasing energy resources, reducing harmful metabolite accumulation, and enhancing antioxidant activity [76]. As an ancient Chinese herbal medicine, *Panax ginseng* C. A. Meyer (*P. ginseng*) has been used as a food or medicine to treat fatigue, and it can exert anti-fatigue effects through antioxidation, anti-inflammatory activity, reduction of metabolite accumulation, or management of energy metabolism [45]. The mechanism of fatigue is very complex, and the different mechanisms also affect each other. There may be new mechanisms that require further study.

## 4. Polysaccharides Ameliorate Fatigue via the Gut-Muscle Axis

### 4.1. Polysaccharide Interferes with Gut Microbiome

One of the tremendous potential targets for affecting host physiology is the gut microbiota. The gut microbiota is composed of microorganisms colonizing the alimentary system, which encompasses approximately trillions of microbes that are highly diverse, complex, and constantly evolving [101,102]. The human body is composed of around 30 trillion cells, which coexist with various microbial communities [103,104]. The biodiversity and overall composition of the gut microbiota plays a crucial role in maintaining normal homeostasis and the long-term health of the host’s body [105]. Bacteria are the most abundant population in the gut microbiota, with more than 1000 different species, consisting mainly of four phyla: *Firmicutes*, *Bacteroidetes*, *Proteobacteria*, and *Actinobacteria* [106]. Probiotics are beneficial bacteria (including *Lactobacillus*, *Bifidobacterium*, *Clostridium butyricum*, etc.), which are largely found in the host’s gastrointestinal tract [107,108]. The composition of gut microbiota varies between individuals and changes dynamically throughout the life cycle, which may be influenced by age, sex, genetics, environment, lifestyle, diet, exercise, and disease conditions [109,110]. Gut microbiota is the most dense and complex micro-ecosystem in the human body, and its functional interactions with the host are critical for host physiology, homeostasis, and sustained health.

The gut microbiota has a crucial role in maintaining human health, and there is an increasing focus on dietary intake to regulate the structure and activity of the gut microecology in order to improve host health, as well as prevent or treat disease [20,111]. One dietary strategy for regulating the microbiota is the consumption of dietary fiber or polysaccharides, which are generally thought to benefit gut health [112]. Most complex carbohydrates and polysaccharides cannot be directly digested by the host, but can be metabolized by microorganisms in the gastrointestinal tract [113]. Over the past few years, investigation of gut microbiota–polysaccharide health has grown exponentially [12,114]. The behavior of polysaccharides in the gut is thought to serve as a “bridge” to the communication between the microbiota and host. Natural polysaccharides participate in the multiple functions of host–microbe symbiosis in the gut, such as promoting the growth and activity of probiotics in the gut. It was reported that *Echinacea purpurea* polysaccharide (EPP) can regulate gut microbial diversity by increasing the abundance of *Bifidobacterium* and *Streptococcus*, while also reducing the abundance of *Prevotella*, *Catenibacterium*, and *Ruminococcus torques* [115]. It is noteworthy that the supplementation of *Eucommiae cortex* polysaccharides (EPs) can remarkably alleviate behavioral disturbances in mice by reducing the ratio of *Firmicutes* to *Bacteroidetes*, inhibiting the expansion of *Escherichia coli*, and promoting the growth of SCFAs-producing bacteria, including *Butyricicoccus*, *Fibrobacter*, and *Roseburia*, in the gut microbiome [116]. Additionally, *Dendrobium fimbriatum* polysaccharide (cDFPW1) restores the stability of the gut microbiome by increasing *Romboutsia* and *Lactobacillus* and decreasing *Parasutterella* and *Acinetobacter* to effectively improve colitis in mice [117]. These studies indicated that polysaccharides can promote the growth of certain beneficial gut bacteria, alter the characteristics of the gut microbiome, and change the physiological conditions of the host, thereby reducing the development of diseases.

In addition to stimulating the growth of certain gut bacteria, dietary polysaccharides can also produce substantial benefits by directly shaping the microbiota [12]. Owing to the differences in the human digestive physiology, non-starch polysaccharides become the main nutrient to reach the microbiome [111]. The polysaccharide from *Moutan Cortex* (MC-Pa) can alleviate hyperglycemia and renal injury in diabetic kidney diseased rats by reconstructing the gut microbiota (mainly affecting phyla and genera), improving intestinal barrier function, and attenuating serum pro-inflammatory mediators [118]. *Gardenia jasminoides Ellis* polysaccharide (GPS) has a protective effect against cholestatic liver injury in mice through ameliorating gut microbiota dysbiosis, protecting the intestinal barrier function integrity, and suppressing the TLR4/NF-κB pathway [119]. Furthermore, EPs can ameliorate obesogenic diet-induced gut dysbiosis by reducing lipopolysaccharide (LPS) concentration and inhibiting the subsequent neuroinflammation, exhibiting potentially immunomodulatory and neuroprotective effects [116]. In fact, gut dysbiosis affects intestinal permeability, systemic inflammation, anabolism, and nutrient availability [120]. Thus, polysaccharides can mediate the changes in gut microbiome composition to influence overall gut health.

Of note, the gut microbiota interaction is the digestion of dietary polysaccharides, which releases host-absorbable energy through fermentation products, and polysaccharides can also promote the production of various active metabolites by the gut microbiota [20,111]. These specific metabolites contribute to suppress inflammation, strengthen the intestinal barrier, activate intestinal immunity, and inhibit pathogens [114]. MC-Pa treatment can improve the contents of SCFAs (acetate, propionate, and valerate), while reducing the concentration of branched-chain fatty acids (BCFAs) (isobutyrate and isovalerate) in the rat cecal to positively affect diabetic kidney disease [118]. EPP not only promotes the proliferation of gut-beneficial bacteria, but also stimulates the beneficial bacteria to metabolize polysaccharides and produce valuable metabolites, including tyrosine, tryptophan, beta-D-fructose 1,6-biphosphate, etc. [115]. There is growing opinion that the metabolites produced by microbes using polysaccharides are key players in gut integrity and host health. Recent evidence proves that GPS can protect the liver by reducing the metabolite levels of bile acids and decreasing the expression of inflammatory factors [119]. Encouragingly, EPs supplementation is effective in improving the cognitive and social behavior of mice via modulating tryptophan and SCFAs metabolism (especially butyrate) in the colon [116]. The correlation analysis revealed a complex relationship between microbiota, metabolites, and host phenotypes [12]. In concert, these results demonstrate that polysaccharides can exert a beneficial modulatory effect on the gut microbiota in particular, leading to a reduced abundance of pathogens, an increased population of favorable bacteria, and the production of SCFAs, supporting the health of the host.

### 4.2. Polysaccharides Ameliorate Fatigue via Gut Microbiota

Recent years have witnessed the increased interest in the gut microbiota as a hotspot of fatigue investigation. Current accumulated data indicate that gut microbiota disturbance can affect the nutritional status, essential metabolism, immune systems, endocrine systems, and neuronal function of the host, and may also affect brain function and behavior, etc. [105,121]. The impairment of intestinal integrity is another extremely important effect of fatigue development [109]. Long-term exhaustive swimming exercise resulted in an increased abundance of *Acetatifactor*, *Allobaculum*, *Oscillibacter*, *Paraprevotella*, and *Roseburia* and a decreased abundance of *Alistips*, *Clostridium*, *Akkermansia*, *Olsenella*, and *Lactobacillus* in mice [122]. The analysis results showed that the beneficial bacteria (*Lactobacillus*, *Akkermansia*, etc.) were decreased and the harmful bacteria (*Candidatus_Planktophila*, *Candidatus_Arthromitus*, etc.) were increased in the guts of mice experiencing weight-loaded swimming fatigue [29]. Moreover, other studies have also shown that exercise fatigue is related to changes in the gut microbiome [22,89]. The deviations in gut microbiota can change the micro-ecological environment of the gastrointestinal tract, linking to the pathogenesis of fatigue and even its complications.

Since the gut microbiota is known to be the causally responsible for the development and deterioration of metabolic syndromes, it may at least partially regulate the occurrence of metabolic imbalance-induced fatigue [113,123]. The elevation of bacteria producing L-lactic acid in the gut microbiome will also lead to the exacerbation of exercise-induced fatigue [124]. The induced exercise stress reduced the level of *Turicibacter* and increased the number of *Ruminococcus gnavus*, which plays a significant role in immune function [63]. In addition, various studies indicated that probiotics can support healthy digestion and resistance function, as well as aid in the regulation of muscle anabolism, the elevation of insulin sensitivity, the increase in amino acid biosynthesis or AMPK signaling, and the limiting of low-grade inflammation and oxidative stress by metabolite production [23,27,125]. The increased relative abundance of *Proteobacteria* is considered as a potential diagnostic criterion for the risk of metabolic disorders and fatigue diseases [126]. Changes in microflora caused by fatigue, such as *Enterococcus* and *Proteobacteria*, which are associated with host oxidative damage and intestinal inflammation [89,109], have been reported. The balanced gut microbiota produces multiple bioactive metabolites that play a marked role in reducing inflammation and ROS production, among other activities [121,127]. The psychological and physical demands during overtraining can trigger stress reactions, activate the sympathetic neural and HPA axis, and lead to the release of central neurotransmitters [52,57,110]. The increase in 5-HT is intimately implicated in the enhanced abundance of intestinal pathogens (*Escherichia*, *Streptococcus*, and *Enterococcus*), which promotes the production of 5-HT [63,128]. Intestinal dysbiosis has been recognized as an emerging factor contributing to fatigue. In general, the gut microbiota could influence fatigue through a variety of mechanisms, implying a close correlation with metabolism, immunity, inflammation, nerve, and HPA axis function. Moreover, the gut microbiota also participates in the regulation of anxiety, emotional stress, cognition, and pain [103,121]. It is well known that the positive effects of dietary additions can dramatically modulate the composition of the gut microbiota to reshape the biological output of gut microbes and regulate physiological status.

Specifically, the anti-fatigue mechanism of these natural polysaccharides may not only be limited to the direct regulation of the target in vivo after absorption, but may also include the pathways used to ameliorate fatigue through regulating gut microbiota (Figure 3). Growing evidence strongly suggests that the anti-fatigue ability of polysaccharides is implicated in the regulation of gut microorganisms, as shown in Table 2. It has been reported that the administration of *Tuber indicum* polysaccharide (TIP) in mice can ameliorate intestinal dysbiosis by decreasing the ratio of *Firmicutes*-to-*Bacteroidetes*, increasing *Bacteroides*, and improving SCFAs concentration and gut integrity to counteract exercise-induced fatigue in mice [126]. Similarly, ethanol-fractional polysaccharides from *Dendrobium officinale* (EPDO) showed the greatest alleviation of fatigue through increasing the proportions of *Bacteroidetes* and *Firmicutes*, enhancing the abundance of *Lactobacillus* and *Bifidobacterium* (known as anti-inflammatory bacteria) in gut microbiome, and accelerating the metabolism in vivo [30]. Moreover, fermented ginseng leaf (FGL) can regulate bacterial diversity (*Bacteroidetes*, *Bacteroidaceae*, *Allobaculum*, and *Akkermansia*) to alleviate exercise-induced fatigue in rats [129]. Thus, the application of polysaccharides can improve the imbalance related to fatigue in the gut ecosystem. Furthermore, the water extraction of ginseng (WEG, containing abundant saccharides, 90.15%, *w/w*) achieves strong therapeutic effects on fatigue in the rat model by reshaping the gut microbial ecosystem and triggering several molecular and cellular signaling pathways (e.g., butyrate, bile acid, or TGR5 signals) [21]. Polysaccharides serve as energy substrates for specific intestinal bacteria that significantly and beneficially regulate gut microbiota. Maca has emerged as a prescription for treating exercise-induced fatigue stemming from its prominent pharmacological properties [48,73,74]. Maca compound prescriptions (MCP) exert anti-fatigue activities through reshaping the gut microbial ecosystem (in phylum, genus, and operational taxonomic units (OTUs) levels) and modulate endogenous metabolism [22,29]. Recent evidence showed that the extracts from *Astragali Radix*, *Codonopsis Radix*, and *Jujubae Fructus* (ACJ) can improve exercise performance by regulating changes in gut metabolites and microbiota to reduce fatigue [89]. Moreover, there was also a certain correlation between fatigue-related biomarkers and the intestinal microbiome [29,89]. For instance, *Odoribacter* and *Roseburia* were positively correlated with the concentration of Na^+^-K^+^-ATPase, Ca^2+^-Mg^2+^-ATPase, and blood glucose, and *Bacteroides* were positively correlated with blood glucose concentration, AMPK expression, but negatively correlated with BLA content [126]. That is, natural polysaccharides play a crucial role in regulating the occurrence of exercise fatigue and the balance of gut microbiota, which may be critical for manual workers or sub-healthy people.

The gut microbiota is closely associated with energy expenditure, systemic inflammation, host metabolism, etc., suggesting that polysaccharides can exert an anti-fatigue effect via the gut microbiota [11,121]. For example, fungal and plant polysaccharides can intervene in nutrient absorption and improve the immune system barrier by ameliorating the intestinal microecological balance [20,114]. Other polysaccharides, such as EPs [116], cDFPW1 [117], MC-Pa [118], and GPS [119], also have a positive effect on different diseases by ameliorating gut microbiota dysbiosis, while simultaneously inhibiting the pro-inflammatory mediators, reducing detrimental metabolites, and enhancing the endogenous antioxidant capacity, which may provide special evidence for their potential anti-fatigue characteristics. Gut microbiota plays an irreplaceable role in the areas of fatigue and fatigue prevention. On the other hand, numerous studies have proven that natural polysaccharides relieve exercise-induced fatigue by mediating multiple mechanisms [68,69,70], but the question remains regarding whether or not they regulate the community structure and function of host microorganism. The complex interactions occurring between polysaccharides, hosts, and different microorganisms are progressively being deciphered. At present, it is generally recognized that exploring the mechanism used by natural polysaccharides to alleviate fatigue, from the perspective of gut microbiota, is a supplement to the existing mechanism research model, and is also the future investigation direction.

### 4.3. Relationship between Gut Microbiota and Muscle

Muscle is the largest organ in human body, with its mass accounting for 40% of the whole-body weight [130]. Skeletal muscles are crucial to human health and disease, and as the largest user of glucose intake and an important storage of body proteins, it is intricately linked with homeostasis, exhibiting important anabolic and catabolic functions [131]. Over the past few decades, the functions and properties of skeletal muscle have been extensively studied, which contributes to the characterization of muscle and provides unique insights for the inter-tissue communication network of the organism [24,49]. Numerous factors contribute to the loss of muscle mass and function, such as senility [26], special diet [27], exercise [46], inflammation [32], mitochondrial dysfunction [16], etc. Another important factor not to be neglected is the gut physiology, as the gut microbiota can interact with skeletal muscle by regulating various processes affecting host physiology, including those involving systemic inflammation, energy provision and use, oxidative stress, and mitochondrial function, as well as endocrine and insulin resistance [23,130]. These factors are known to conspicuously influence muscle state over a long period of time, especially gut microbiota, while gut dysbiosis is associated with lower muscle mass and poorer physical function.

There may be a homeostatic equilibrium between the gut microbiota and skeletal muscle. Skeletal muscles are physiologically far away from the gut, but the signals arising from gut microbiome interactions constitute the link between gut microbiota activity and muscle through systemic mechanisms [23,24,28]. There was a significant modification of muscle glucose homeostasis by gut microbiota, as it mediates the regulation of SCFAs, glucose transporters G protein-coupled receptor 41 (Gpr41), sodium-glucose cotransporter 1 (Sglt1), and muscle glycogen to affect the skeletal muscle endurance and metabolic function of the host [132]. In addition, maintaining healthy mitochondrial quality and function plays a vital role in terms of muscle contraction and stress responses [28,88]. The presence of gut microbiota can significantly improve muscle quality, enhance insulin-like growth factor 1 (IGF-1) expression, reduce markers of muscle atrophy, improve the antioxidant capacity and mitochondrial function of muscles, and maintain the normal progress of energy metabolism, thereby reversing skeletal muscle impairment [133]. Intestinal ecological disorders, as a pathological link, participate in the dysregulated autophagic process and mitochondrial damage, which are dramatically associated with the maintenance of skeletal muscle quality [125]. The increased production of the gut bacteria SCFAs can positively affect skeletal muscle mass and body function in human beings [25]. Perturbations in the composition and function of microbiota are connected with autoimmune disorders. For example, the relative abundance of *Clostridium* and total SCFAs in patients with myasthenia gravis was decreased [134]. The balanced gut microbiota produces a variety of bioactive metabolites that play an important role in reducing inflammation and ROS production, as well as in mitigating damage to the myocytes [130]. It has been suggested that the crosstalk between gut microbiota and skeletal muscle plays a role in different pathological conditions, such as chronic inflammation of the gut [132]. Patients with chronic intestinal inflammation exhibit inadequate intestinal barrier function, enhanced abundance of *Escherichia coli*, and overexpression of pro-inflammatory cytokines (interleukin-1 (IL-1), TNF-α, etc.), which may affect the quality of skeletal muscle and lead to sarcopenia [28]. And, the probiotics that can limit muscle disease or promote host health performance are mainly *Lactobacilli* and *Bifidobacteria* [23]. Gastrointestinal and intestinal microbiota are believed to be closely related to diet, muscle function, and metabolism [135]. Taken together, these studies strongly support the conclusion that gut bacteria are necessary for optimal skeletal muscle function and mobility in the host.

### 4.4. Impact of the Gut-Muscle Axis on Muscle Movement Ability

The connection between muscle and gut microflora was termed the “gut-muscle axis”. Currently, the presence of a gut-muscle axis that regulates the quality and function of skeletal muscle has been proved in animal models and in a small number of human experiments [26,136]. Previous studies advocated that the relative representation of *Faecalibacterium prausnitzii*, *Roseburia inulinivorans*, and *Alistipes shahii* was depleted in sarcopenic subjects, in which their metabolic capacity for producing SCFAs, carotenoids, isoflavones, and amino acids was reduced [26]. Recent publications have highlighted that vitamin D supplementation can improve the health of obese women through the gut-muscle axis, mainly improving the intestinal microbiome function and homeostasis, as well as facilitating muscle anabolism and maintaining muscle mass [27]. The effect of the gut-muscle axis is manifested by the significant reversal of muscular atrophy in mice by treatment with intestinal microorganisms and their hallmark metabolites [133]. Emerging evidence suggests that probiotics (*Lactobacillus casei Shirota*) could enhance muscle function via the gut-muscle axis, reduce the declines in muscle strength, improve mitochondrial function, and maintain SCFAs (acetic, isobutyric, butyric, and hexanoic acid) levels in the aged mice [136]. The gut-muscle axis describes how the gut microbiota impacts muscle quality and function by influencing body metabolism and SCFAs levels.

The loss of skeletal muscle quality and function will lead to a series of adverse consequences, beginning with the body’s locomotion mechanisms [46,130]. In sports, muscles are subjected to intense efforts and repeated (slight) injuries, and are always required to recover from the injury [23]. The gut microorganism plays a fundamental role in modulating muscle adaptation to extensive training [105]. Increased probiotics in the gut can restore imbalances in the intestinal ecosystem, normalize serum biomarkers, and reduce intestinal inflammation, thereby relieving fatigue and enhancing exercise function [122]. Exercise training is beneficial to intestinal health, suggesting that exercise can regulate the composition of the microbial community and increase its biodiversity [24,131]. Physical activity levels can alter biodiversity or alter the level of specific bacterial taxa in established intestinal microbiota [137]. Surprisingly, regular physical exercise may help combat the decrease in the levels of butyrate-producing microbiota (*Clostridiales*, *Roseburia*, and *Lachnospiraceae*) and *Lactobacillus*, which commonly occurs in aging [24]. Moreover, regular exercise can reduce inflammation in the body’s circulation and promote good health by altering gut microbiota [125]. In recent studies, moderate-intensity treadmill exercise obviously enhanced the athletic performance of mice and enhanced the relative abundance of *Bifidobacteria* and *Coprococcus*, which was associated with the elevated metabolism of glucose, flavonoids, arginine, and proline [138]. Exercise is associated with increased biodiversity and representation of taxa with beneficial metabolic functions. Conversely, training to exhaustion may also disrupt intestinal homeostasis, which is associated with elevated intestinal barrier permeability and enhanced oxidative stress; it may also promote inflammation and negative metabolic consequences [120]. Although investigation regarding the understanding of exercise and intestinal microorganisms has developed rapidly in recent years, questions remain concerning the health effects of microflora changes caused by different levels of exercise. Furthermore, there was a high correlation between physical and emotional stress during exercise and changes in gastrointestinal microbiota composition [63]. In summary, the human gut microbiota is essential for maintain muscle mass and normal exercise, in terms of movement mechanisms.

Indeed, an increasing number of studies have provided evidence that exercise capacity can be modulated by the gut-muscle axis. The gut-muscle axis actually modulates muscle protein deposition and muscle function and repairs immune system damage during exercise [120]. A properly balanced microbiome may have positive impacts on muscle protein synthesis, mitochondrial biogenesis, and muscle glycogen storage, and may also alleviate injuries caused by excessive exercise by reducing ROS production and inflammatory markers [105]. Moreover, exercise can interfere with the interaction of the gut-muscle axis to improve muscle quality and function [131]. The gut-muscle axis may participate in the exercise process through a variety of mechanisms, including accelerating glucose metabolism from dietary nutrients, regulating immunity, increasing mitochondrial function, and reducing inflammation [136,139,140]. In conclusion, the effect of the gut-muscle axis in human pathophysiology may be bidirectional, with gut microbiota affecting muscle and exercise performance, and conversely, exercise contributes to the reshaping of the composition of the microbiota [120,131]. In consequence, it seems important to optimize and regulate the action of the gut-muscle axis, which participates in a variety of potential responses of the body.

### 4.5. Polysaccharide Interactions with the Gut-Muscle Axis

Polysaccharides play an important role in intestinal microbiology [12,20,111]. Previous investigation showed that each individual’s response to polysaccharides primarily depends on their baseline intestinal microbiota, but whether or not intestinal microbiota directly drive the gut-muscle axis function during polysaccharide intake is unclear [115]. Carbohydrate intake, especially indigestible polysaccharide consumption, is the preferred substrate for intestinal bacteria [105]. In the intestine, microbial components and metabolites promote the fermentation of indigestible polysaccharides into SCFAs, such as acetate, propionate, and butyrate [141]. It is well known that glycogen metabolism is essential for muscle function and high-intensity exercise [142]. Polysaccharides may affect muscle status during exercise through microbial action. Previous studies showed that mice treated with *Paecilomyces hepialid* spores (HPS) and mycelium (HPM) (contains polysaccharides) showed less skeletal muscle fiber spacing and breakage, restored metabolic levels, and regulate gut homeostasis during fatigue [122]. Because glycogen is the key energy substrate for long-term exercise, the regulation of muscle availability through intestinal microbiota represents an effective mechanism that contributes to the gut-muscle axis [132]. In this scenario, long-term adjusted nutritional strategies have already been formulated to favor muscle health or limit loss by supplementing dietary fiber and polysaccharides, which also modulate the microbiota [12,112]. Furthermore, the AMPK and PGC-1α signaling pathways are implicated in the gut-muscle axis [125]. Dysbiosis could reduce the level of p-AMPK in the skeletal muscles and the liver and inhibit the AMPK signaling pathway [27]. Known as important regulators of cellular metabolic homeostasis, the AMPK and PGC-1α pathways play a key role in autophagy activity, inflammation reaction, and insulin resistance in skeletal muscle tissues [83,84]. Some polysaccharides also effectively activated the AMPK and PGC-1α pathway, which suppress inflammation and oxidative stress to achieve anti-fatigue effects [68,78,98]. However, the relationship between polysaccharides and the gut-muscle axis remains unclear and requires further investigation.

### 4.6. Polysaccharides Relieve Fatigue by Gut-Muscle Axis

Because the gut-muscle axis pathway is associated with muscle quality and exercise ability, it is hypothesized that the occurrence and treatment of exercise-induced fatigue may occur via gut-muscle axis signaling. The ability of polysaccharides to mediate the action of gut microorganisms and play a positive role in exercise fatigue has been widely studied [29,30,126]. Zheng et al. demonstrated that FGL relieves exercise-induced fatigue via regulating the gut microbiota–muscle axis, and the Spearman analysis showed that the gut microbiota may affect muscle metabolism by regulating muscular activity [129]. To date, since few direct studies have explored the treatment of exercise fatigue by dietary polysaccharide mediation of the gut-muscle axis, such a summary will be helpful to gain insight into fatigue treatments (Figure 4). A growing number of researchers are investigating the benefits of probiotics in improving exercise performance and validating the role of the gut-muscle axis [143,144] by considering the bacteria species associated with the this axis, such as *Bifidobacteria*, *Lactobacillus*, *Faecalibacterium prausnitzii*, *Escherichia coli*, etc. [24,28,145]. Some studies have shown that exercise-induced fatigue tends to be associated with lower levels of *Lactobacillus* and *Bifidobacterium* [29,122], but the addition of some polysaccharides can increase their relative abundance, improve muscle mass and exercise performance, and reduce exercise fatigue [30,115]. On the other hand, EPs supplementation limits the abundance of *Escherichia coli*, which is overexpressed in inflammatory patients and damages muscles [28,116]. And there are lower levels of *Faecalibacterium prausnitzii* in sarcopenia [26]. It seems that bacteria beneficial for muscle fitness are negatively correlated with exercise fatigue. Moreover, PI3K-Akt-mTOR and myostatin/activin signaling may be involved in the gut-muscle axis by inhibiting the NF-κB and FOXO signaling pathway [145]. Overall, the gut microbiota is deeply involved in metabolism and inflammation in both direct and indirect ways, influencing host health [125]. Some polysaccharides can regulate the above effects [29,32,98]. However, the effects of polysaccharides on gut microbiota composition and metabolic function are highly dependent on the target population [112]. In summary, in the future, further studies on humans and animals are required to continue to disentangle the complex relationships between polysaccharides, the gut-muscle axis, and exercise-induced fatigue.

## 5. Conclusions and Perspectives

Natural polysaccharides, which are non-toxic to human health and rarely cause side effects, are widely used as protective ingredients in research regarding biological effects. Polysaccharides have been widely used in the development of health foods used to provide fatigue prevention effects. It has been reported that some polysaccharides play important roles in exercise-induced fatigue, but the molecular mechanism of their anti-fatigue effects is still unclear. This review summarized and analyzed the molecular mechanisms of dietary polysaccharides in alleviating fatigue, as discovered in recent years, serving to deepen our understanding of the anti-fatigue mechanisms of polysaccharides. The occurrence and development of fatigue is complex, and it is associated with multiple mechanisms. Regulating the balance between gut microbiota and muscle function is a key molecular mechanism for polysaccharides to alleviate exercise-induced fatigue. The polysaccharide-mediated gut-muscle axis is an integration and innovation of these two mechanisms. Classical anti-fatigue research typically focused on the ability of polysaccharides to directly regulate energy metabolism, reduce the accumulation of metabolites, improve immune function and antioxidant activity, inhibit the inflammatory response, and regulate the autonomic nervous and endocrine systems. The regulation of the gut-muscle axis provides a novel direction for exploring the molecular mechanism of dietary polysaccharides in alleviating exercise fatigue. However, there is still a scarcity of evidences that polysaccharides relieve exercise fatigue through gut-muscle axis regulation. Further research on the mechanism of polysaccharide, gut microbiota, and muscle function will help to elucidate the molecular mechanism of polysaccharide anti-fatigue properties.

Due to the significant correlation and complexity between polysaccharide structure and its biological activities, it is necessary to establish a detailed structure–function correlation network for polysaccharides. In addition, the following questions must be considered: What structural features of polysaccharides have the strongest anti-fatigue activity? What structural changes can affect their anti-fatigue activity? Do polysaccharides with the same structure also exhibit similar potential anti-fatigue effects? These issues need further clarification in the future research. In fact, there are still many issues that need to be resolved in these fields. Polysaccharides are very complex substances, and some researchers believe that polysaccharides can directly enter cells or be absorbed into the bloodstream, while most investigators believe that polysaccharides can only exert their biological functions by altering the gut microbiota and its metabolites. However, to date, pharmacokinetics studies of polysaccharides have not been systematically reported in the literature. In addition, future research on the anti-fatigue effects of polysaccharides could further explore the impacts of fecal microbiota transplantation and track the pharmacokinetics of gut microbiota metabolites, which will be enhance our understanding of the anti-fatigue mechanisms of polysaccharides. All of these questions need to be answered in future research; this will facilitate the efficient utilization of polysaccharides.

Lastly, it is important to emphasize that some additional issues should also be considered when investigating the mechanisms of anti-fatigue polysaccharides, for example, how do polysaccharides regulate signal pathways in cells? Which transcription factors can be activated by polysaccharides? How do polysaccharides participate in gene transcriptions of microRNAs, circRNAs, long-coding RNAs, and other noncoding genes? All of these questions are still in the initial stages of exploration, and investigating them will provide theoretical guidance for the development of more effective functional anti-fatigue polysaccharides. Overall, exploring the dietary polysaccharides that exert anti-fatigue functions via gut-muscle axis regulation remains a daunting task.

## Figures and Tables

**Figure 1 foods-12-03083-f001:**
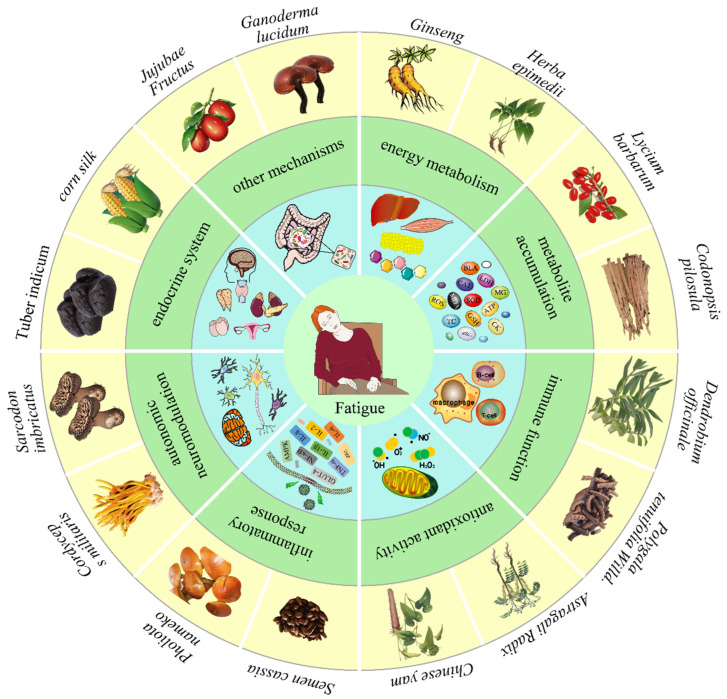
The anti-fatigue activity of dietary polysaccharide. Dietary polysaccharides alleviate fatigue, primarily by influencing energy metabolism, reducing metabolite accumulation, improving immune function, enhancing antioxidant activity, inhibiting the inflammatory response, interfering with autonomic neuromodulation, modulating the endocrine system, regulating gut microbiota homeostasis, etc.

**Figure 2 foods-12-03083-f002:**
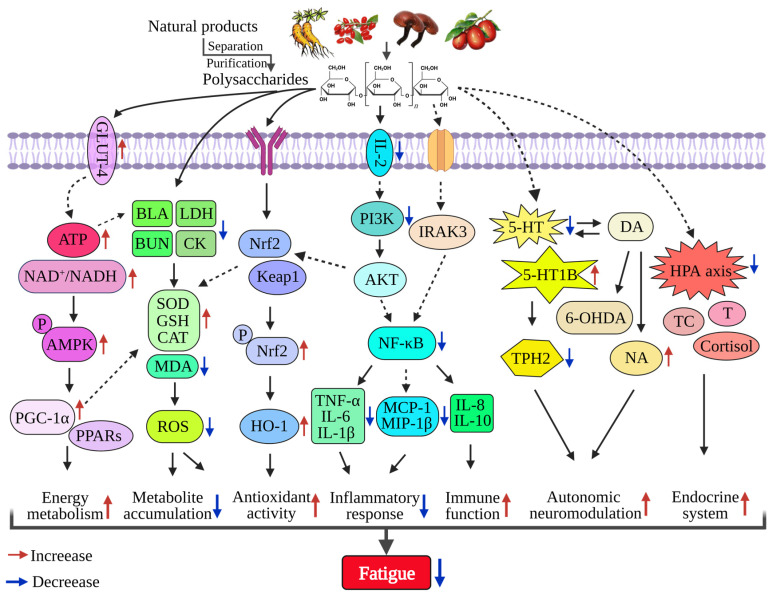
The molecular mechanism of dietary polysaccharides in alleviating fatigue. Dietary polysaccharides mainly alleviate fatigue through the following pathways: regulating energy metabolism mediated by the ATP and AMPK/PPARα/PGC-1α signaling pathways; regulating metabolic balance by reducing the content of BLA, LDH, and BUN, acting on CAT, SOD, GSH, and other kinases and inhibiting ROS; enhancing antioxidant activity mediated by the Nrf2/Keap1/HO-1 signaling pathways; inhibiting the PI3K/AKT and NF-κB signaling pathways mediating the expression of pro-inflammatory cytokines (TNF-α, IL-6, IL-1β, etc.); activating autonomic neuroprotection by regulating 5-HT, DA, NA, etc.; and modulating the endocrine system mediated by the HPA axis.

**Figure 3 foods-12-03083-f003:**
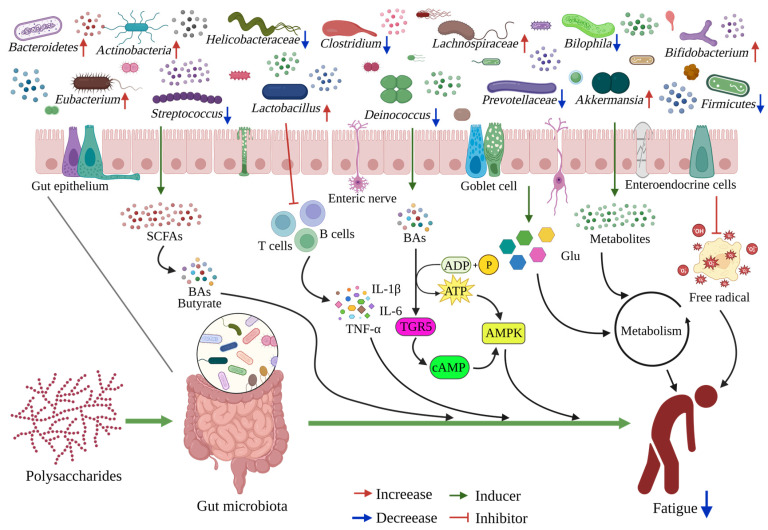
The schematic diagram of dietary polysaccharides alleviating fatigue through the gut microbiota. Most dietary polysaccharides are utilized in the large intestine by microbial fermentation, and some metabolites will have certain effects on the gut microbiota and contents. Dietary polysaccharides can gradually increase the beneficial bacteria and short-chain fatty acids, inhibiting harmful bacteria and their metabolites. The metabolites produced by gut microbiota are transported through the intestinal epithelial cells, and they inhibit the inflammatory response, promote energy and carbohydrate metabolism, and inhibit oxidative stress, and they are also involved in regulating host health and producing anti-fatigue effects.

**Figure 4 foods-12-03083-f004:**
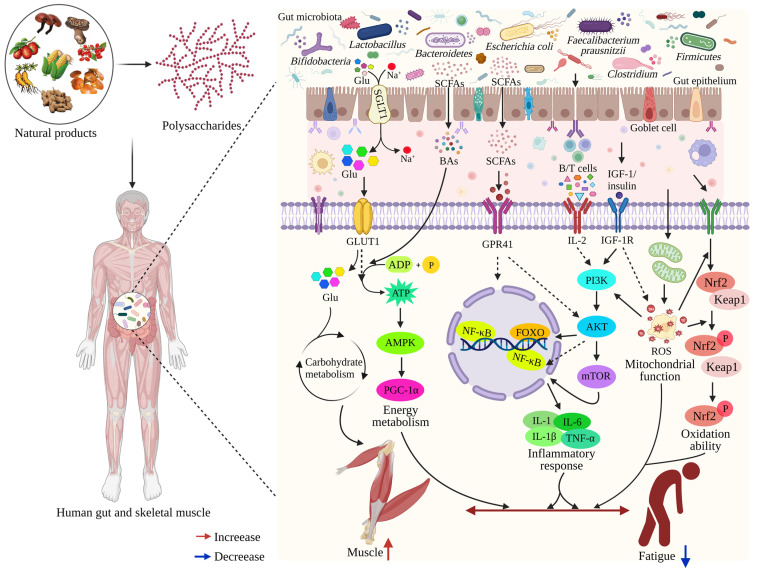
The schematic representation of dietary polysaccharides alleviating fatigue through the gut-muscle axis. The gut-muscle axis is a two-way communication between the muscle and the digestive tract. The microbiota can act on the muscles throughout the body via the microbe–gut-muscle axis. Microorganisms in the gut produce some metabolites (SCAFs, etc.) under the action of dietary polysaccharides. Some metabolites pass directly through the intestinal epithelial cells and will act on the muscle tissues and cells, directly or indirectly, through blood circulation, which will cause the cells to undergo physiological and biochemical reactions, subsequently exerting a certain impact on fatigue. Supplementation with dietary polysaccharides, by acting on gut microbiota and its metabolites, indirectly activates the AMPK/PGC-1α, PI3K/AKT, NF-κB, and Nrf2/Keap1 signaling pathways to regulate energy metabolism, reduce inflammation levels, enhance mitochondrial function and antioxidant capacity, and further maintain muscle mass and function, thus playing a role in alleviating fatigue.

**Table 1 foods-12-03083-t001:** The monosaccharide composition and the anti-fatigue mechanisms of dietary polysaccharides.

Sources	Polysaccharides	Total Sugar Content	Monosaccharide Composition	Preparation	Model/ Number	Method/Dose/ Time	Characteristics	Ref.
corn silk	PCS	58.6%	Rha, Ara, Xyl, Man, Glu, and Gal; ratio is 0.17:0.30:0.26:0.35: 1.00:0.57	Water-extracting, alcohol precipitating, and ultrasonic-assisted	Kunming mice, (*n* = 20)	oral gavage; 50, 100, 200, 400 mg/kg; 4 weeks	body weight, food consumption, weights of main organ →; swimming time to exhaustion ↑; LDH, HG, GLU, TC, Cr ↑; BUN, LA, ALT, ALP ↓; AST, TG →	[71]
*Herba epimedii*	HEP2-a	60.1%	Man, Rha, Glu, Gal, Ara, and galacturonic acid in percentages of 4.41%, 5.43%, 31.26, 27.07%, 23.43%, 8.40%	soaked in 95% alcohol, extracted with distilled water at 80 °C	CFS rat, (*n* = 10)	oral gavage; 50, 100, 200 mg/kg; 4 weeks	body weight, number of standing ↑; time needed to locate the platform; motionless time in the tail-suspension test ↓; noradrenaline, creatine ↑	[72]
*Dendrobium officinale*	DOP	30%	/	/	inbred strain male BALB/c mice, (*n* = 8)	oral; 50 mg/kg; 30 days	endurance, body weight, food intake ↑; LDH, BUN, MDA, CK, TG, and LD in serum ↓; serum SOD/GSH-Px and gastrocnemius glycogen ↑; cell variability of T and B lymphocytes ↑	[41]
*Lepidium* *meyenii Walp.* (Maca)	MP	99.2%	D-GalA: D-Glc: L-Ara: D-Man: D-Gal: L-Rha = 35.07:29.98:16.98:13.01:4.21:0.75 (mol, %)	extracted with hot water at 90 °C	ICR mice, (*n* = 20)	Intragastrically; 25, 50, 100 mg/kg bw/d; 30 days	viscera indices, body weight gain →; swimming durations, average swimming speeds ↑; GSH-Px, CK ↑; LDH, BUN, MDA, LA ↓	[73]
*Lepidium* *meyenii Walp.* (maca)	MPS-1, MPS-2	93.2%; 91.5%	MPS-1: xylose, arabinose, galactose, glucose, with the mole ratio 1:1.7:3.3:30.5; MPS-2: arabinose, galactose, glucose, with the mole ratio 1:1.3:36.8	extracted with distilled water at 80 °C	male Kunming mice, (*n* = 10)	oral gavage; 20, 100 mg/kg·d; 30 days	exhausting swimming time ↑; LG ↑; BLA, BUN, LDH ↓	[74]
*Epidium meyenii Walp*. (Maca)	ME	19.72%	/	extracted in boiling water	ICR mice, (*n* = 10); H_2_O_2_-induced C2C12 cell 6 h	oral; 10 mL per kg bw; 4 weeks; 0.10 mg mL^−1^	grip-strength, exercise time in the rotarod test ↑; BLA, BUN, LDH, ROS ↓; skeletal muscle damage ↓; NAD^+^, NADH ↑; no cytotoxicity, cytoprotective and 68.8% anti-oxidation effects, ROS ↓; mitochondrial quality ↑	[16]
*Phragmites rhizoma*	PEP	85.6%	/	hot water decoction	male ICR mice, (*n* = 6)	oral; 1 g/kg; 10 days	body weight →; time to reach fatigue, swimming endurance capacity ↑; glucose, TG, T. chol., GSH, SOD, CAT, GPx ↑; LDH, CK, cortisol, MPO, TBARS ↓	[75]
*Codonopsis pilosula*	POL	15.52%	/	extracted with 95% ethanol and decocted with boiling water	male ICR mice, (*n* = 8)	oral gavage; 0.25, 0.5, 1.0 g/kg; 21 days	swimming time, hypoxia tolerance ↑; MG, LG, GSH ↑; BUN, LDH, MDA ↓	[76]
*Polygala tenuifolia* Willd.	/	/	/	extracted with 95% ethanol reflux and 67 °C water	male mice, (*n* = 10)	gavage, 0.1, 0.2, 0.4 mg/g, 30 days	weight-loaded swimming time ↑; LG, MG, LDH ↑; BLA, BUN ↓; scavenging rate of hydroxyl and DPPH free radical ↑	[77]
*Chinese yam (Dioscorea opposita Thunb.)*	CYPs	73.20%	rhamnose, glucuronic acid, glucose, galactose, and arabinose with a molar ratio of 0.01:0.06:1.00:0.17:0.01	water-extracting, alcohol-precipitating	Swiss mice, (*n* = 10)	oral; 100 mg/kg; 14 days	exhausting swimming time, movement distance, body weight ↑, static time ↓; ATP, SOD ↑; IL-lβ, MDA, BUN, LDH ↓; tumor volume and tumor weight ↓	[18]
*Semen cassiae*	SCP	5.42%.	/	extracted using aqueous solvent (water)	BALB/c male mice	oral; 100 mg/kg; 4 weeks	swimming duration, body weight, food intake ↑; BUN, CK, TG, LA, LDH, MDA ↓; SOD, GPX, liver and muscle glycogen ↑; lymphocyte proliferation ↑	[14]
*Ginseng*	WSGP-S3	8.01%	Rha, GalA, Glc, Gal, Ara GlcA was 35.48, 6.98, 5.98, 32.02, 16.95, 2.67 mol%	extracted under reflux extraction with 95% ethanol; washed with distilled water, and extracted with hot water	male Kunming mice, (*n* = 10)	oral; 25, 50, 75 mg/kg; 30 days	body weight →; thymus and spleen index ↑; exhaustive swimming time↑; LG, MG, SOD, CAT, GSH-Px ↑; BUN, LA, MDA ↓; T or B lymphocyte proliferation ↑	[17]
*Panax ginseng C. A. Meyer*	APS-1	96%	Man, Rha, GlcA, GalA, Glc, Gal and Ara with an area ratio of 3: 4:2:17:7:40:27	extracted with boiling water and separated by DEAE-52 cellulose ion exchange column and gel column chromatography	male C57BL/6J mice, (*n* = 8)	Intragastric; 50, 100, 150 mg/kg; 15 days	fatigue tolerance time ↑; BLA, LDH, BUN, MDA ↓; SOD, CAT, CK ↑; LKB1, p-AMPK, PGC-1α and Glut4 ↑	[78]
*Lycium barbarum*	LBP1-SeNPs	80.1%	arabinose (24.0%), ylose (25.0%), glucose (20.2%), galactose (23.1%)	immersed in 95% ethanol and extracted with ultra-pure water at 90 °C	male ICR mice, (*n* = 20)	Intragastric; 0.5, 2, 4 Se/kg/d; 30 days	body weigh →; exhaustive swimming time ↑; liver and muscle glycogen, SOD ↑; BUN, BLA, MDA ↓	[79]
*Lycium barbarum*	LBP	4.73%, 97.22%	Rha, Ara, Gal, Glu, Xyl and Man with a molar ratio of 1.00:11.35:6.10:0.56:1.08:0.71	extracted with water, and DEAE-52 cellulose was used for purification	4-week-old male SD rats, (*n* = 8)	Gavage; 120, 360 mg/kg/d; 28 days	exhaustive swimming time ↑; LA and CK, serum glucose, ATP and glycogen ↑	[68]
*Inonotus obliquus*	PIO-1	73.2%	mannose, glucose, galactose, xylose, and arabinose with the molar ratio of 1.0:1.9:3.5:18.5:5.7	hot water extraction	male Kunming mice, (*n* = 10)	oral; 50 mg/kg/day; 30 days	climbing duration, swimming time ↑; immobility time ↓; BLA, BUN, LDH, 5-HT ↓; GRAF1 ↑	[19]
*Sarcodon imbricatus*	SI	35.22%	/	/	Kunming male mice, (*n* = 10)	oral gavage; 0.25, 0.5, 1 g/kg; 20 days	exercise tolerance↑, immobility in the tail suspension test ↓; glycogen, ATP, SOD, GSH-Px ↑, LD, BUN, ROS, MDA ↓; Nrf2, SOD1, SOD2, HO-1, and CAT ↑	[44]
*Ganoderma lucidum*	GLPs	/	Fuc, Ara, Gal, Glc, Xyl, Man	extracted by hot water	male Balb/C rats, (*n* = 10)	Gavage; 100, 150, 200 mg/kg; 4 weeks	body weight →; exercise time ↑; BUN, CK, MDA, BLA ↓; Gly, TG, GPx, SOD, LDH ↑; fat transformation ↑	[15]
*Pholiota nameko*	PNP	82%	/	hot water extracting-alcohol precipitating	adult Kun Ming mice, (*n* = 6)	gavage; 150, 300, 450 mg/kg; 15 days	forced swim time ↑; LDH, ATPase, SOD ↑; MDA ↓	[80]
*Cordyceps militaris*	CMPB	36.81%	Man (44.06%), Gal (28.41%), Glc (7.42%), Rha (4.98%), Xyl (4.93%), Ara (4.42%)	extracted, isolated and purified via DEAE-cellulose 52 and Sepharose CL-6B columns	male ICR mice, (*n* = 15)	Intragastric; 400, 800 mg/kg; 5 weeks	DPPH, superoxide anion, hydroxyl radicals ↑; stick rotation time, learning and memory abilities ↑; LD, LDH, BUN, MDA, ROS ↓; SOD, GSH-Px, eNOS, Glu/GABA, BDNF, PI3K, Nrf2, HO-1 ↑	[69]
*Spirulina platensis*	PSP	70.13%	/	/	SD male rats, (*n* = 10)	Intragastric; 50, 100, 200 mg kg^−1^; 7 days	exhaustion time ↑; Hb, 5-HT1B ↑; LA, BUN, CK, 5-HT, TPH2 ↓	[81]
*Chitosan oligosaccharide*	COS	/	/	/	SD rats, (*n* = 6)	Intragastric; 200 mg/kg; 4 weeks	spleen weight ↑; LAC, the ratio of T cell/CD8^+^ T cell ↑; ALB, BUN, blood creatinine, lymphocytes, LDH, CREA, TNF, IL-2 ↓; TC, TG, HDL, Na^+^, K^+^, and Ca^2+^, platelet, IL-10, IL-4 →	[82]

Note: ↑ indicates increase; ↓ indicates decrease; → indicates no change.

**Table 2 foods-12-03083-t002:** The changes in gut microbiota after the administration of anti-fatigue polysaccharides.

Sources	Polysaccharides	Total Sugar Content	Monosaccharide Composition	Preparation	Model/ Number	Method/Dose/ Time	Changed Abundance of Gut Microbiota and SCFAs	Characteristics	Ref.
Ginseng	WEG	90.15%	Glc, Gal, Ara, GlcA, GalA, Man, Rha, and Fuc, with the molar ratios of 1588:96:80:94:79:21:13:1	prepared by hot water extraction	SD rats, (*n* = 8)	intraperitoneal injection; 1.42 g kg^−1^; 14 days	TUDCA, TCDCA, UDCA 3-sulfate, CDCA 3-sulfate ↓; butyrate ↑; tryptophan, LPC ↑; *Bacteroidetes*, *Lactobacillus*, *Bacteroides* ↑, *Firmicutes* ↓; *Bifidobacterium*, *Coprococcus* ↑; *Anaerotruncus*, *Streptococcus*, *Blautia*, *Clostridium* ↓	body weight ↑; LA, CP, LDH, MDA, BUN, TG ↓; GPx, Glc, SOD ↑; IL-1β ↓, TGR5 ↑	[21]
*Lepidium meyenii Walp.* (Maca)	MCP	34.78 mg/mL	/	extracted in boiling water (*w/v*, 1:8)	ICR mice, (*n* = 10)	oral, 1, 2, 4 g/kg w; 30 days	*Lactobacillus*, *Akkermansia*, *Lachnospiraceae*, *Bacteroides*, *Blautia*, *Roseburia*, *Parabacteroides*, *Clostridia_UCG-014*, *Marvinbryantia* ↑; *Candidatus_Planktophila*, *Candidatus_Arthromitus* ↓	grip-strength, rotarod ↑; blood sugar, NAD(H), MG, ATP ↑; BLA, BUN, LDH, ROS ↓	[29]
Fermented ginseng leaf	FGL	19.2 mg/g	/	fermented by *S. cerevisiae*	male SD rats, (*n* = 12)	intragastric administration; 10, 20, 50 mg/kg/d; 4 weeks	acetic acid, propionic acid, n-buryric acid, i-butyric acid, total SCFAs in feces ↑; *Bacteroidetes*, *Bacteroidetes*/*Firmicute* ratio, *Allobaculum*, *Akkermansia*, *Bifidobacterium*, *Eubacterium* ↑; *Verrucomicrobiales*, *Coprobacillus*, *Erysipelotrichaceae* ↓	endurance time ↑; hypoxanthine, isoprostane ↓; TNF-α, IL-6, IL-10 ↓; SOD, GSH-Px ↑; MDA ↓; MyHC-I, MyHC-II ↓; Pax7, MyoD1 ↑	[129]
Tuber indicum	TIP	90.25%	Rha, Man, Glu, and Gal, with a molar percentage of 6.73%, 13.36%, 76.78% and 3.13%	extracted with ultrapure water, and precipitated with of 100% ethanol	male C57BL/6 mice, (*n* = 8)	Gavage; 0.1, 0.3, 0.9 g/kg body weight; 40 days	acetic acid, propionic acid, n-butyric acid, i-butyric acid, total SCFAs ↑; *Firmicutes*, *Proteobacteria* ↓; *Bacteroidetes*, *Actinobacteria* ↑; *Prevotellaceae*, *Ruminococcaceae*, *Helicobacteraceae* ↓; *Porphyromonadaceae*, *Bacteroidaceae*, *Rikenellaceae* ↑; *Odoribacter*, *Bacteroides*, *Roseburia*↑; *Deferribacteres*, *Lachnospiraceae* →	exhaustive swimming time, weight, feed efficiency ↑; LG, MG, BG ↑, BLA, BUN, CK ↓; LDH, Na^+^K^+^-ATPase, Ca^2+^Mg^2+^-ATPase ↑; p-p38/p38, p-AMPK/AMPK ↓; occluding ↑, ZO-1 →	[126]
*Astragali Radix*, *Codonopsis Radix*, *Jujubae Fructus*	ACJ	43.46 mg/g	/	extracted with water, concentrated under vacuum in a rotary evaporator	male ICR mice, (*n* = 11)	oral gavage; 4.69, 18.75 g kg^−1^ d^−1^; 6 weeks	*Actinobacteria*, *Proteobacteria*, *TM7*, *Cyanobacteria* ↑; *Oscillospira*, *Hymenobacter*, *Deinococcus*, *Bilophila*, *Streptococcus* ↓; *Adlercreutzia*, *Fructobacillus*, *Desulfovibrio* ↑	Body weight →; exhaustive swimming time ↑; BUN, LA ↓; LG ↑	[89]
*Dendrobium officinale*	EPDO	92.6%	Man and Glc ratio is 2.0–4.0	immersed with 80% ethanol, extracted with deionized water	male ICR mice, (*n* = 6)	oral gavage; 100, 150, 200 mg/kg; 4 weeks	*Bacteroidetes*, *Actinobacteria* ↑; *Firmicutes*, *Proteobacteria* ↓; *Lactobacillus*, *Bifidobacterium*, *Roseburia* ↑; *Acidaminoccus* ↓	body weight →; swimming time ↑; LG, MG →; SOD ↑; BLA, BUN, MDA ↓	[30]

Note: ↑ indicates increase; ↓ indicates decrease; → indicates no change.

## Data Availability

The data used to support the findings of this study can be made available by the corresponding author upon request.

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
