# Peer review of "Dietary Polysaccharides Exert Anti-Fatigue Functions via the Gut-Muscle Axis: Advances and Prospectives"

_foods, 2023, doi:10.3390/foods12163083_

Round 1
Reviewer 1 Report
General characterization of the manuscript
The reviewed manuscript is related to specific aspects of the general problem of the use of polysaccharides in food science and nutrition. Natural dietary polysaccharides are known to have specific anti-fatigue effects with minor side-effects and rich pharmacological activities, which could help develop potential anti-fatigue agents. Gut-muscle axis as a newly proposed concept, its still unclear causal relationship, and the gut microbiota as a potential intervention target for the prevention and treatment of muscle-related diseases is also discussed. There are several points to be addressed by the authors.
Proposed corrections to manuscript
1. Lines 119-122: a long sentence at these lines should be better divided into two sentences "It has been demonstrated that prolonged endurance exercise exacerbates the production of reactive oxygen species (ROS)." and "Although ROS are necessary for normal force production in muscle, high levels of ROS appear to promote systolic dysfunction [42,43].". Or alternatively the two parts of long sentence should have a meaningful connection.
2. Line 128: "Inflammation is thought to be the most important molecular mechanism of fatigue." That is loosely related to Lines 140-141 "The sequence of peripheral theories leading to exercise induced fatigue can be schematically denoted as: energy exhaustion > metabolite accumulation > immunoregulation > oxidative stress > inflammation." May be, an arrow symbol or a mathematical symbol "<" instead of a symbol ">" was assumed to be included in this sequence? Alternatively the sequence of significance of different theories, from high to low, seems to be presented here.
3. Line 150: " (i.e. serotonergic, 5-HT), dopamine ...". Obviously, a word "neurotransmitter", "central neurotransmitter", or similar, should be included in parentheses.
4. Line 184: "signaling involving different tissues, such as leptin, adiponectin, and ghrelin". But leptin, adiponectin, and ghrelin are not tissues but hormones. Adipose tissue is known to be a highly active endocrine organ secreting a range of hormones. leptin, ghrelin, adiponectin, and resistin [e.g., Meier & Gressner, 2004]. Besides, "a fine interaction of signaling" (the more so a signaling-interacting counterpart is not pointed out in this sentence) should be also rephrased.
5. Line 190: the abbreviation "SD" is not spelled out throughout the text and Abbreviations list (Lines 915-948). At a first appearance in the text (Line 190), there should be "Sprague-Dawley (SD) rats".
6. Table 1: monosaccharide composition of corn silk, "Arab" should be replaced by "Ara".
7. Table 1: the evidence of the order of polysaccharides positions in the leftmost column of the table is not clear. There is a mixed arrangement of plants, algae, ascomycetes fungi, basidiomycetes fungi, and others. It would be very useful to make the corresponding subdivision of the column.
8. Line 238: the abbreviation "SeNPs" is not spelled out throughout the text and Abbreviations list (Lines 915-948). At a first appearance in the text (Line 238), there should be "LBP1-decorated selenium nanoparticles (LBP1-SeNPs)".
9. Line 356: "hydrogenase oxidase (CAT)" should be changed. Hydrogenases are hydrogen-activating enzymes. Hydrogenases catalyse the reversible oxidation and production of molecular hydrogen. Catalase (CAT) catalyze the breakdown of hydrogen peroxide H2O2 into water and oxygen. Thus, hydrogenase is not CAT.
10. Line 357: "catalyze ROS reaction" should be changed.
11. Lines 361-364: "polysaccharide of Polygala tenuifolia ... scavenge the hydroxyl and DPPH free radical". One could conclude from this phrase that DPPH is one of the harmful free radicals, along with e.g. hydroxyl radical, existing in a body. That is why the authors should differentiate between a model stable analytically applied radical DPPH, and the processes in vivo described as the consequences of Polygala tenuifolia polysaccharide impact.
12. Lines 364-370: "APS-1 ... could alleviate oxidative damage caused by MDA, and thus ..." should be rephrased. MDA is not a direct reason for oxidative damage. Malondialdehyde (MDA) is a byproduct of lipid peroxidation process, and MDA production in vivo is used for analytical purposes to evaluate a lipid peroxidation state.
13. Line369: "ME can ... regulate NAD+/NADH to exert anti-oxidative damage" should be rephrase ("damage").
14. Line 372: "(polysaccharide – commented by reviewer) LBP plus aerobic exercise can regulate the genes of ...", along with the item at Line 463 "polysaccharides may also regulate some genes", and the item at Line 469: "alterations in some genes ... can also be used to evaluate fatigue" should be all replaced by more strict definitions.
15. Line 420: "The yeast beta-glucan (Saccharomyces cerevisiae) (YBG)" should be replaced by "The yeast (Saccharomyces cerevisiae) beta-glucan (YBG)".
16. Line 429: "oxidation/oxidation resistance system" is not-very-appropriate phrase used solely in the work [Bai et al., 2023], and is desirable to be reconstructed.
17. Line 541: the abbreviation "LPS" is not spelled out throughout the text and Abbreviations list (Lines 915-948). At a first appearance in the text (Line 541), there should be "lipopolysaccharide (LPS)", and a short note is desirable why LPS concentration has to be reduced.
18. Line 555: "valuable metabolites ... beta-D-fructose 1,6-bisphosphate , etc." should be replaced by "valuable metabolites ... beta-D-fructose 1,6-biphosphate , etc." (without "s" in biphosphate) to discriminate between this metabolite and enzymes as D-fructose-1, 6-bisphosphate aldolase, fructose-1, 6-bisphosphatase, etc.
19. Line 730: "capacity of producing SCFAs was reduced, including carotenoid, isoflavone, and amino acid" should be better replaced by "capacity of producing SCFAs, carotenoids, isoflavones, and amino acids was reduced" so as to avoid the attribution of carotenoids, isoflavones, and amino acids to short-chain fatty acids.
20. Line 758: here again (see analogs at point 19) "metabolism of glucose, flavonoid, arginine, and proline", however, in contrast to glucose, arginine, and proline, flavonoid is not a separate compound, but a whole chemical class. Therefore, "flavonoids" should be used, or a definite flavonoid with its chemical name mentioned.
21. The English level of the manuscript should be improved, especially at Lines 56, 64, 109-110, 111-112, 119-122, 130, 134, 148, 149-152, 153-154, 159, 181, 258, 297, 307, 324, 371-372, 390, 392, 414-416, 421, 451, 459, 466, 476, 536, 570-571, 615-618, 793, 824-826.
References used to review the manuscript
Meier, U., Gressner, A.M. Endocrine regulation of energy metabolism: review of pathobiochemical and clinical chemical aspects of leptin, ghrelin, adiponectin, and resistin. Clinical Chemistry 2004, 50(9), 1511–1525. doi:10.1373/clinchem.2004.032482
Bai, L., Tan, C., Ren, J., Liu, J., Zou, W., Liu, G., & Sheng, Y. Cordyceps militaris acidic polysaccharides improve learning and memory impairment in mice with exercise fatigue through the PI3K/NRF2/HO-1 signalling pathway. International Journal of Biological Macromolecules 2023, 227, 158–172. doi:10.1016/j.ijbiomac.2022.12.071

The English level of the manuscript should be improved, especially at Lines 56, 64, 109-110, 111-112, 119-122, 130, 134, 148, 149-152, 153-154, 159, 181, 258, 297, 307, 324, 371-372, 390, 392, 414-416, 421, 451, 459, 466, 476, 536, 570-571, 615-618, 793, 824-826.
Author Response
Response to Reviewer 1
General characterization of the manuscript
The reviewed manuscript is related to specific aspects of the general problem of the use of polysaccharides in food science and nutrition. Natural dietary polysaccharides are known to have specific anti-fatigue effects with minor side-effects and rich pharmacological activities, which could help develop potential anti-fatigue agents. Gut-muscle axis as a newly proposed concept, its still unclear causal relationship, and the gut microbiota as a potential intervention target for the prevention and treatment of muscle-related diseases is also discussed. There are several points to be addressed by the authors.
Reply: Thanks for the reviewer’s positive comments for the manuscript. According with your comments, we amended the relevant parts of the manuscript one by one. We carefully checked every sentence of the manuscript, revised some sentences or paragraphs, and corrected every mistake (labeled in red). I hope our revision can meet the requirements of reviewers and Foods.
Proposed corrections to manuscript
Question 1. Lines 119-122: a long sentence at these lines should be better divided into two sentences "It has been demonstrated that prolonged endurance exercise exacerbates the production of reactive oxygen species (ROS)." and "Although ROS are necessary for normal force production in muscle, high levels of ROS appear to promote systolic dysfunction [42,43].". Or alternatively the two parts of long sentence should have a meaningful connection.
Reply: Thanks for the reviewer comments. We revised the long sentence as two sentences "Low levels of reactive oxygen species (ROS) are necessary for the generation of normal muscle strength, but high concentrations of ROS can lead to impaired muscle contractility [42]." and "Studies indicated that long endurance exercise could produce large amounts of ROS, which affected the exercise ability of muscle and caused muscle fatigue [43].".
Question 2. Line 128: "Inflammation is thought to be the most important molecular mechanism of fatigue." That is loosely related to Lines 140-141 "The sequence of peripheral theories leading to exercise induced fatigue can be schematically denoted as: energy exhaustion > metabolite accumulation > immunoregulation > oxidative stress > inflammation." May be, an arrow symbol or a mathematical symbol "<" instead of a symbol ">" was assumed to be included in this sequence? Alternatively the sequence of significance of different theories, from high to low, seems to be presented here.
Reply: Thanks for the valuable comments. We have revised the "Inflammation is thought to be the most important molecular mechanism of fatigue." as "Inflammation is thought to be another important molecular mechanism of fatigue."
We have revised it to "The sequence of peripheral theories leading to exercise induced fatigue can be schematically denoted as: energy exhaustion → metabolite accumulation → immunoregulation → oxidative stress → inflammation." in our resubmitted manuscript.
Question 3. Line 150: " (i.e. serotonergic, 5-HT), dopamine ...". Obviously, a word "neurotransmitter", "central neurotransmitter", or similar, should be included in parentheses.
Reply: Thanks for the comments. We have revised it to "Within fatigue investigation most emphasis has been given to neurotransmitters released by neuron, including 5-hydroxtryptamine (5-HT), dopamine (DA), noradrenaline (NA), gamma-aminobutyric acid (GABA), and acetylcholine, etc., which can alter central fatigue and motoneuronal output [50]." in our resubmitted manuscript.
Question 4. Line 184: "signaling involving different tissues, such as leptin, adiponectin, and ghrelin". But leptin, adiponectin, and ghrelin are not tissues but hormones. Adipose tissue is known to be a highly active endocrine organ secreting a range of hormones. leptin, ghrelin, adiponectin, and resistin [e.g., Meier & Gressner, 2004]. Besides, "a fine interaction of signaling" (the more so a signaling-interacting counterpart is not pointed out in this sentence) should be also rephrased.
Reply: Thanks so much for your question. I apologize for the incorrect description in the manuscript, we have made detailed revisions to this sentence in the resubmitted manuscript.
"The CNS controls the regulation of energy balance through signal transduction, including leptin, adiponectin, ghrelin, etc., and these signal transduction molecules provide important feedback for the hypothalamus to regulate energy homeostasis [43].".
Question 5. Line 190: the abbreviation "SD" is not spelled out throughout the text and Abbreviations list (Lines 915-948). At a first appearance in the text (Line 190), there should be "Sprague-Dawley (SD) rats".
Reply: Thank you very much! We have revised the first occurrence of "SD" in the text to "Sprague-Dawley (SD)", added it to the Abbreviations list, and made corresponding modifications to the "SD" throughout the text in our resubmitted manuscript.
Question 6. Table 1: monosaccharide composition of corn silk, "Arab" should be replaced by "Ara".
Reply: Thanks. We have revised the "Arab" to "Ara" in Table 1.
Question 7. Table 1: the evidence of the order of polysaccharides positions in the leftmost column of the table is not clear. There is a mixed arrangement of plants, algae, ascomycetes fungi, basidiomycetes fungi, and others. It would be very useful to make the corresponding subdivision of the column.
Reply: Thanks for the reviewer’s comments. The leftmost column in Table 1 indicates the sources of polysaccharides, which are mainly from plants (corn silk, Maca, Ginseng, Lycium barbarum, etc.), a small amount from fungi (Sarcodon imbricatus, Inonotus obliquus, Ganoderma lucidum, Pholiota nameko, Cordyceps militaris), algae (Spirulina platensis), and Chitosan oligosaccharide. The order of the columns is primarily based on the year of the cited references, starting from 2017 to 2023.
We have revised the Table 1 according to the requirements of the reviewer. The order of the polysaccharide sources in the leftmost column was adjusted to plants, fungi, algae, and Chitosan oligosaccharide, from most to least.
Question 8. Line 238: the abbreviation "SeNPs" is not spelled out throughout the text and Abbreviations list (Lines 915-948). At a first appearance in the text (Line 238), there should be "LBP1-decorated selenium nanoparticles (LBP1-SeNPs)".
Reply: Thanks for the comments. We have revised the first occurrence of " LBP1-decorated SeNPs (LBP1-SeNPs)" in the text to "LBP1-decorated selenium nanoparticles (LBP1-SeNPs)" and made corresponding modifications in the Abbreviations list.
Question 9. Line 356: "hydrogenase oxidase (CAT)" should be changed. Hydrogenases are hydrogen-activating enzymes. Hydrogenases catalyse the reversible oxidation and production of molecular hydrogen. Catalase (CAT) catalyze the breakdown of hydrogen peroxide H2O2 into water and oxygen. Thus, hydrogenase is not CAT.
Reply: Thanks, we modified "hydrogenase oxidase (CAT)" to "catalase (CAT)" in our resubmitted manuscript.
Question 10. Line 357: "catalyze ROS reaction" should be changed.
Reply: Thanks reviewer. According to your suggestion, we have deleted the "catalyze ROS reaction" and modified the sentence to "The presence of antioxidant enzymes such as catalase (CAT), glutathione peroxidase (GSH-Px), and superoxide dismutase (SOD) promotes oxidative defense, scavenges free radicals and reduces oxidative damage [92]." in our resubmitted manuscript.
Question 11. Lines 361-364: "polysaccharide of Polygala tenuifolia ... scavenge the hydroxyl and DPPH free radical". One could conclude from this phrase that DPPH is one of the harmful free radicals, along with e.g. hydroxyl radical, existing in a body. That is why the authors should differentiate between a model stable analytically applied radical DPPH, and the processes in vivo described as the consequences of Polygala tenuifolia polysaccharide impact.
Reply: Thanks for the comments. According with your comments, we amended the relevant sections in our resubmitted manuscript.
"For example, polysaccharide of Polygala tenuifolia Willd. can decrease the concentrations of BLA and BUN, and increase the levels of LG, muscle glycogen (MG), and LDH in exhaustive exercise mice; it has high scavenging rates of hydroxyl free radical and DPPH free radical in vitro and exhibits good antioxidant properties [94]."
Question 12. Lines 364-370: "APS-1 ... could alleviate oxidative damage caused by MDA, and thus ..." should be rephrased. MDA is not a direct reason for oxidative damage. Malondialdehyde (MDA) is a byproduct of lipid peroxidation process, and MDA production in vivo is used for analytical purposes to evaluate a lipid peroxidation state.
Reply: Thanks for the reviewer’s comments. According to your suggestion, we have removed the incorrect description of "alleviate oxidative damage caused by MDA" and revised the sentence to "It has been proven that APS-1 could reduce the accumulations of BLA, LDH, BUN, MDA and increase the activities of SOD, CAT, and CK, which could prolong the fatigue tolerance time of mice [75]." in our resubmitted manuscript.
Question 13. Line369: "ME can ... regulate NAD+/NADH to exert anti-oxidative damage" should be rephrase ("damage").
Reply: Thank you very much! We revised the sentence to "Furthermore, it has been suggested that the ME can abrogate ROS accumulation, up-regulate NAD+/NADH to reduce exercise-induced metabolic stress, and prevent oxidative stress-induced damage [16]." in our resubmitted manuscript.
Question 14. Line 372: "(polysaccharide – commented by reviewer) LBP plus aerobic exercise can regulate the genes of ...", along with the item at Line 463 "polysaccharides may also regulate some genes", and the item at Line 469: "alterations in some genes ... can also be used to evaluate fatigue" should be all replaced by more strict definitions.
Reply: Thanks for the reviewer comments. According with your comments, we have provided more stricter definitions for some of the gene representation mentioned above in our resubmitted manuscript.
The detailed modifications are as follows:
We revised the sentence from "LBP plus aerobic exercise can regulate the genes of fatty acid synthesis and oxidation in the liver of SD rats, activate AMPK, and increase the expression of PPARα and its co-activator PGC-1α [78]." to "LBP plus aerobic exercise can regulate the synthesis and oxidation of liver fatty acids in SD rats by activating AMPK and increasing the expression of PPARα and its co-activator PGC-1α [78].".
We revised the sentence from "Moreover, polysaccharides may also regulate some genes to resist fatigue." to "Moreover, polysaccharides may also regulate some other fatigue-related genes, such as GRAF1, BDNF, and LKB1.".
We revised the sentence from "Therefore, alterations in some genes related to metabolism, nervous, and other can also be used to evaluate fatigue." to "Therefore, to some extent, alterations in these genes can also be used to evaluate fatigue.".
Question 15. Line 420: "The yeast beta-glucan (Saccharomyces cerevisiae) (YBG)" should be replaced by "The yeast (Saccharomyces cerevisiae) beta-glucan (YBG)".
Reply: Thanks, we have modified "The yeast beta-glucan (Saccharomyces cerevisiae) (YBG)" to "The yeast (Saccharomyces cerevisiae) beta-glucan (YBG)" in our resubmitted manuscript.
Question 16. Line 429: "oxidation/oxidation resistance system" is not-very-appropriate phrase used solely in the work [Bai et al., 2023], and is desirable to be reconstructed.
Reply: Thanks reviewer. According to your suggestion, we have reconstructed the sentence to "Excessive exercise can lead to physical fatigue, disrupt the balance of the oxidation/antioxidant system in the body, and thereby damage the CNS [69]." in the resubmitted manuscript.
Question 17. Line 541: the abbreviation "LPS" is not spelled out throughout the text and Abbreviations list (Lines 915-948). At a first appearance in the text (Line 541), there should be "lipopolysaccharide (LPS)", and a short note is desirable why LPS concentration has to be reduced.
Reply: Thanks reviewer. We have revised the first appearance of "LPS" in the text to "lipopolysaccharide (LPS)" and added it to the Abbreviations list.
We modified "Furthermore, EPs can ameliorate obesogenic diet-induced gut dysbiosis, reduce lipopolysaccharide (LPS) concentration, and inhibit neuroinflammation, exhibit potential immunomodulatory and neuroprotective effects [116]" to "Furthermore, EPs can ameliorate obesogenic diet-induced gut dysbiosis by reducing lipopolysaccharide (LPS) concentration and inhibiting the subsequent neuroinflammation, exhibit potential immunomodulatory and neuroprotective effects [116]" in our resubmitted manuscript.
Question 18. Line 555: "valuable metabolites ... beta-D-fructose 1,6-bisphosphate, etc." should be replaced by "valuable metabolites ... beta-D-fructose 1,6-biphosphate, etc." (without "s" in biphosphate) to discriminate between this metabolite and enzymes as D-fructose-1, 6-bisphosphate aldolase, fructose-1, 6-bisphosphatase, etc.
Reply: Thanks reviewer, this is a good idea! According to the comments, we modified the "beta-D-fructose 1,6-bisphosphate" to "beta-D-fructose 1,6-biphosphate " in our resubmitted manuscript.
Question 19. Line 730: "capacity of producing SCFAs was reduced, including carotenoid, isoflavone, and amino acid" should be better replaced by "capacity of producing SCFAs, carotenoids, isoflavones, and amino acids was reduced" so as to avoid the attribution of carotenoids, isoflavones, and amino acids to short-chain fatty acids.
Reply: Thank you for good suggestion. According to your suggestion, we revised the sentence to "Previous investigation advocated that the relative representation of Faecalibacterium prausnitzii, Roseburia inulinivorans, and Alistipes shahii was depleted in sarcopenic subjects, which their metabolic capacity of producing SCFAs, carotenoids, isoflavones, and amino acids was reduced [26]." in our resubmitted manuscript.
Question 20. Line 758: here again (see analogs at point 19) "metabolism of glucose, flavonoid, arginine, and proline", however, in contrast to glucose, arginine, and proline, flavonoid is not a separate compound, but a whole chemical class. Therefore, "flavonoids" should be used, or a definite flavonoid with its chemical name mentioned.
Reply: Yes, thanks for the reviewer comments. We revised the sentence to "In recent studies, moderate-intensity treadmill exercise obviously enhanced athletic per-formance of mice, and enhanced the relative abundance of Bifidobacteria and Coprococcus, which was associated with elevated metabolism of glucose, flavonoids, arginine, and proline [138]." in our resubmitted manuscript.
Question 21. The English level of the manuscript should be improved, especially at Lines 56, 64, 109-110, 111-112, 119-122, 130, 134, 148, 149-152, 153-154, 159, 181, 258, 297, 307, 324, 371-372, 390, 392, 414-416, 421, 451, 459, 466, 476, 536, 570-571, 615-618, 793, 824-826.
Reply: Thanks for the reviewer’s comments. We are very sorry for the grammatical, spelling, and detail errors in our manuscript. According with your comments, we amended the relevant parts of the manuscript one by one. We carefully checked every sentence of the manuscript, corrected every mistake, and revised the English and language of some sentences or paragraphs (labeled in red).
I hope our revision can meet the requirements of reviewers and Foods.

Reviewer 2 Report
Review report
The article titled "Dietary Polysaccharides Exert Anti-Fatigue Functions via Gut- 1 Muscle Axis: Advance and Prospectives" is very well written, structurally organized and deserve the possible publication.
Here are some suggestions which need to be addressed
1. The conclusion section needs improvement. The first line of the conclusion is grammatically incorrect.
2. Overall, minor English corrections are require.
Minor English editing is required.
Author Response
Response to Reviewer 2
Comments and Suggestions for Authors
Review report
The article titled "Dietary Polysaccharides Exert Anti-Fatigue Functions via Gut- 1 Muscle Axis: Advance and Prospectives" is very well written, structurally organized and deserve the possible publication.
Reply: Thank you very much! Thanks for the reviewer’s positive comments and contributions to the manuscript.
Thanks and best regards!
Here are some suggestions which need to be addressed
Question 1. The conclusion section needs improvement. The first line of the conclusion is grammatically incorrect.
Reply: Thank you for good suggestion. I am sorry for the grammatical errors. According with your comments, we have revised the "Conclusion" section in our resubmitted manuscript.
We have modified the first sentence of the "Conclusion" section to "Natural polysaccharides are widely used as protective ingredients in biological effects research, which are non-toxic to human health and rarely cause side effects.".
Question 2. Overall, minor English corrections are require.
Reply: Thanks for the reviewer’s comments. I am sorry for the grammatical, spelling, and detail errors. According with your comments, we amended the relevant parts of the manuscript one by one. We carefully checked every sentence of the manuscript, corrected every mistake, and revised the English and language of some sentences or paragraphs (labeled in red).

Reviewer 3 Report
In general terms, this is a interesting and well-written manuscript. I have some comments to improve the document:
The most basic concepts were left out. It is not explained what a polysaccharide is, what chemical characteristics it has, from what type of monomers it is formed, among others.
Chemistry is totally left out in this review, even though different reactions between molecules are constantly being named.
Sentence in lines 56-57 is not understandable (it’s linear or branched biopolymers of numerous monosaccharides connected by glycosidic bonds, with high structural diversity and complexity). Please re-write it.
Sentence in lines 73-75 is not clear (Therefore, some natural polysaccharides play an active role in exercise and muscle-related fatigue, and whether this effect mediates gut-muscle axis?) If you are trying to express a research question, please be more direct and specific.
Please include a reference for the following affirmations:
(lines 179-180): “The direct current stimulation can cause transient or long-term changes in excitability of cortical neuronal pathways and improve the occurrence and development of fatigue.”
(lines 289-293): “AMPK and PGC-1α are the key regulatory factors for poly-289 saccharides to participate in energy metabolism.”
(lines 504-506): “Gut microbiota is the most dense and complex micro-ecosystem possessed by the human body, its functional interactions with the host are critical for host physiology, homeostasis, and sustained health.”
I suggest including a table of abbreviations.
Write “It has been” instead of “It’s” in line 304
Write “It is” instead of “It’s” in line 427
Minor corrections should be done.
Author Response
Response to Reviewer 3
Comments and Suggestions for Authors
In general terms, this is a interesting and well-written manuscript. I have some comments to improve the document:
The most basic concepts were left out. It is not explained what a polysaccharide is, what chemical characteristics it has, from what type of monomers it is formed, among others.
Chemistry is totally left out in this review, even though different reactions between molecules are constantly being named.
Reply: Thanks for the reviewer’s positive comments. According to your comments, we have further revised the relevant parts of the manuscript. I hope our revision can meet the requirements of reviewers and Foods.
According to your suggestion, we added the basic concepts of polysaccharide in the "Introduction". "Polysaccharide is one of the crucial building blocks of cell structure; it is a biopolymer composed of many monosaccharides connected by glycosidic bonds, with structural diversity and complexity, no sweetness and insoluble in water [12].".
In this manuscript, we mainly focus on the anti-fatigue effects and possible molecular mechanisms of dietary polysaccharides, and systematically elaborated the polysaccharides exert anti-fatigue effects through the gut microbiota and gut-muscle axis.
Proposed corrections to manuscript
Question 1. Sentence in lines 56-57 is not understandable (it’s linear or branched biopolymers of numerous monosaccharides connected by glycosidic bonds, with high structural diversity and complexity). Please re-write it.
Reply: Yes, thanks for the reviewer comments. We revised the sentence to "Polysaccharide is one of the crucial building blocks of cell structure; it is a biopolymer composed of many monosaccharides connected by glycosidic bonds, with structural diversity and complexity [12]." in our resubmitted manuscript.
Question 2. Sentence in lines 73-75 is not clear (Therefore, some natural polysaccharides play an active role in exercise and muscle-related fatigue, and whether this effect mediates gut-muscle axis?) If you are trying to express a research question, please be more direct and specific.
Reply: It is good comment! We revised the sentence to "Therefore, it is possible that some natural polysaccharides play an active role in fatigue by mediating the gut-muscle axis." in our resubmitted manuscript.
Question 3. Please include a reference for the following affirmations:
(lines 179-180): “The direct current stimulation can cause transient or long-term changes in excitability of cortical neuronal pathways and improve the occurrence and development of fatigue.”
(lines 289-293): “AMPK and PGC-1α are the key regulatory factors for poly-289 saccharides to participate in energy metabolism.”
(lines 504-506): “Gut microbiota is the most dense and complex micro-ecosystem possessed by the human body, its functional interactions with the host are critical for host physiology, homeostasis, and sustained health.”
Reply: Thanks for the reviewer comments. Through our careful examination of these sentences in the manuscript, these sentences are a summary of the contents for the corresponding paragraphs and their cited references. Therefore, the reference was not cited again.
Question 4. I suggest including a table of abbreviations.
Reply: Thank you very much! In our resubmitted manuscript, we have added a list of "Abbreviations" in front of the list of "References".
Question 5. Write “It has been” instead of “It’s” in line 304
Reply: Thanks for the comments. We revised the "It’s" to "It has been" in our resubmitted manuscript.
Question 6. Write “It is” instead of “It’s” in line 427
Reply: Thanks, we revised the "It’s" to "It is", and we checked and corrected the similar errors in the resubmitted manuscript.
